# Activity in perceptual classification networks as a basis for human subjective time perception

Warrick Roseboom [1,2], Zafeirios Fountas [3], Kyriacos Nikiforou [3], David Bhowmik[3], Murray Shanahan[3,4] & Anil K. Seth [1,2,5]

Despite being a fundamental dimension of experience, how the human brain generates the perception of time remains unknown. Here, we provide a novel explanation for how human time perception might be accomplished, based on non-temporal perceptual classification processes. To demonstrate this proposal, we build an artificial neural system centred on a feed-forward image classification network, functionally similar to human visual processing. In this system, input videos of natural scenes drive changes in network activation, and accumulation of salient changes in activation are used to estimate duration. Estimates produced by this system match human reports made about the same videos, replicating key qualitative biases, including differentiating between scenes of walking around a busy city or sitting in a cafe or office. Our approach provides a working model of duration perception from stimulus to estimation and presents a new direction for examining the foundations of this central aspect of human experience.

[1] Department of Informatics, University of Sussex, Falmer, Brighton BN1 9QJ, UK. [2] Sackler Centre for Consciousness Science, University of Sussex, Falmer, Brighton BN1 9QJ, UK. [3] Department of Computing, Imperial College London, London SW7 2RH, UK. [4] DeepMind, London N1C 4AG, UK. [5] Canadian Insitutute for Advanced Research (CIFAR), Azrieli Programme on Brain, Mind, and Consciousness, Toronto, ON, Canada. Correspondence and requests for materials should be addressed to W.R. (email: wjroseboom@gmail.com)

In recent decades, predominant models of human time perception have been based on the presumed existence of neural processes that continually track physical time—so-called pacemakers—similar to the system clock of a computer[1–3]. Clear neural evidence for pacemakers at psychologically relevant timescales has not been forthcoming and so alternative approaches have been suggested (see refs. [4–8]). The leading alternative proposal is the network-state-dependent model of time perception, which proposes that time is tracked by the natural temporal dynamics of neural processing within any given network[9–11]. While recent work suggests that network-state-dependent models may be suitable for describing temporal processing on short time scales[11–13], such as may be applicable in motor systems[10,13–15], it remains unclear how this approach might accommodate longer intervals (greater than a few seconds) associated with subjective duration estimation.

In proposing neural processes that attempt to track physical time as the basis of human subjective time perception, both the pacemaker and state-dependent network approaches stand in contrast with the classical view in both philosophical[16] and behavioural work[17–19] on time perception that emphasises the key role of perceptual content, and most importantly changes in perceptual content, in subjective time. It has often been noted that human time perception is characterised by its many deviations from veridical perception[20–23]. These observations pose substantial challenges for models of subjective time perception that assume subjective time attempts to track physical time precisely. One of the main causes of deviation from veridicality lies in basic stimulus properties. Many studies have demonstrated the influence of stimulus characteristics such as complexity[17,24] and rate of change[25–27] on subjective time perception, and early models in cognitive psychology emphasised these features[17,28–30]. Subjective duration is also known to be modulated by attentional allocation to tracking time (e.g. prospective versus retrospective time judgements[31–35] and the influence of cognitive load[33,34,36]).

Attempts to integrate a content-based influence on time perception with pacemaker accounts have hypothesised spontaneous changes in clock rate (e.g. ref. [37]), or attention-based modulation of the efficacy of pacemakers[31,38]. No explicit efforts have been made to demonstrate the efficacy of state-dependent network models in dealing with these issues. Focusing on pacemaker-based accounts, assuming that content-based differences in subjective time are produced by attention-related changes in pacemaker rate or efficacy implies a specific sequence of processes and effects. First, it is necessary to assume that content alters how time is tracked, and that these changes cause pacemaker/accumulation to deviate from veridical operation. Changed pacemaker operation then leads to altered reports of time specific to that content. In contrast to this approach, we propose that the intermediate step of a modulated pacemaker, and the pacemaker in general, be abandoned altogether. Instead, we propose that changes in perceptual content can be tracked directly in order to determine subjective time. A challenge for this proposal is that it is not immediately clear how to quantify perceptual change in the context of natural ongoing perception. However, recent progress in machine learning provides a solution to this problem.

Accumulating evidence supports both the functional and architectural similarities of deep convolutional image classification networks (e.g. ref. [39]) to the human visual processing hierarchy[40–42]. Changes in perceptual content in these networks can be quantified as the collective difference in activation of neurons in the network to successive inputs, such as consecutive frames of a video. We propose that this simple metric provides a sufficient basis for subjective time estimation. Further, because this metric is based on perceptual classification processes, we hypothesise that the produced duration estimates will exhibit the same content-related biases as characterise human time perception. To test our proposal, we implemented a model of time perception using an image classification network[39] as its core and compared its performance to that of human participants in estimating time for the same natural video stimuli.

Our results show that model estimates closely match human estimates, including overestimation of shorter and underestimation of longer durations. The match between human- and model-produced estimates improves when the model input is constrained to approximate human visual-spatial attention (based on human gaze data). Finally, model-produced duration estimates replicate the same pattern of biases by video scene type (city, country, office/cafe) as human duration estimates for these same scenes. Overall, these results support our proposal that human-like time estimation can be generated based on tracking changes in perceptual content, as measured in the dynamics of perceptual classification networks, providing a new approach to understand this fundamental aspect of human experience.

## Results

**Human-like time estimation based on perceptual classification.** The stimuli for human and model experiments were videos of natural scenes, such as walking through a city or the countryside, or sitting in an office or cafe (see Supplementary Movie 1; Fig. 1c). These videos were split into durations between 1 and 64 s and used as the input from which our model would produce estimates of duration (see Methods for more details). To validate the performance of our model, we had human participants watch these same videos and make estimates of duration using a visual analogue scale (Fig. 1). Participants' gaze position was also recorded using eye tracking while they viewed the videos.

The videos were input to a pre-trained feed-forward image classification network[39]. To estimate time, the model measured whether the Euclidean distance between successive activation patterns within a given layer, driven by the video input, exceeded a dynamic threshold (Fig. 2). The dynamic threshold was implemented for each layer, following a decaying exponential corrupted by Gaussian noise and resetting whenever the measured Euclidean distance exceeded it. For a given layer, when the activation difference exceeded the threshold, a salient perceptual change was determined to have occurred, and a unit of subjective time was accumulated (see Supplementary Fig. 4 and Supplementary Discussion for model performance under a static threshold). To transform the accumulated, abstract temporal units extracted by the model into a measure of time in standard units (seconds) for comparison with human reports, we trained support vector regression to estimate the duration of the videos based on the accumulated salient changes across network layers. Importantly, the regression was trained on the physical durations of the videos, not the human-provided estimates. Therefore, an observed correspondence between model- and human-produced estimates would demonstrate the ability of the underlying perceptual change detection and accumulation method to model human duration perception, rather than the more trivial task of mapping human reports to specific videos/durations (see Methods for full details of system design and training).

We initially had the model produce estimates under two input scenarios. In one scenario, the entire video frame was used as input to the network. In the other, input was spatially constrained by biologically relevant filtering—the approximation of human visual-spatial attention by a "spotlight" centred on real human gaze fixation. The extent of this spotlight approximated an area equivalent to human parafoveal vision and was centred on the participants' fixation measured for each time-point in the video.

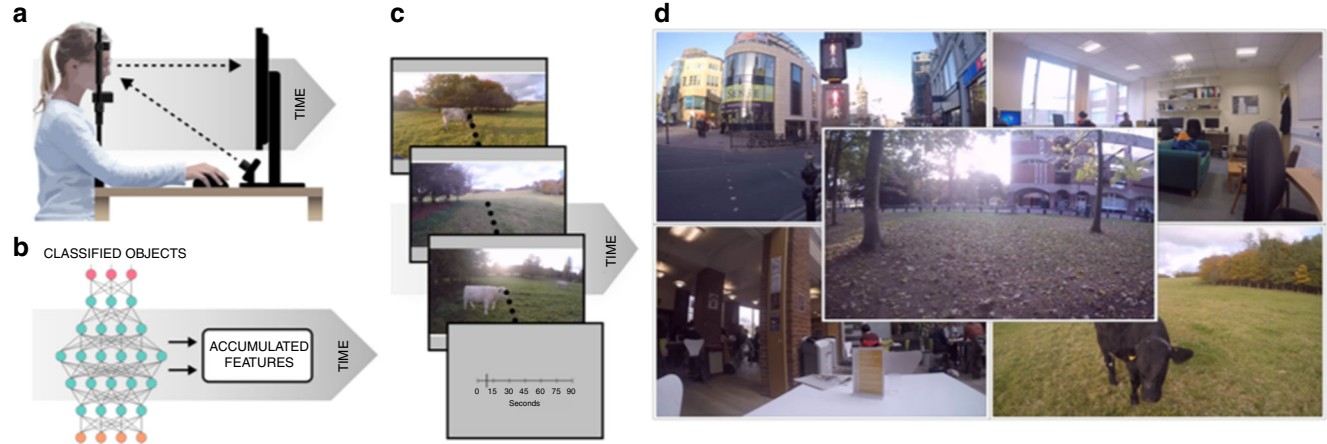

**Fig. 1** Experimental apparatus and procedure. **a** Human participants observed videos of natural scenes and reported the apparent duration while we tracked their gaze direction. **b** Depiction of the high-level architecture of the model used for simulations (see also Fig. 2 below). **c** Frames from a video used as a stimulus for human participants and input for simulated experiments. Human participants provided reports of the duration of a video in seconds using a visual analogue scale. **d** Videos used as stimuli for the human experiment and input for the model experiments included scenes recorded walking around a city (top left), in an office (top right), in a cafe (bottom left), walking in the countryside (bottom right) and walking around a leafy campus (centre)

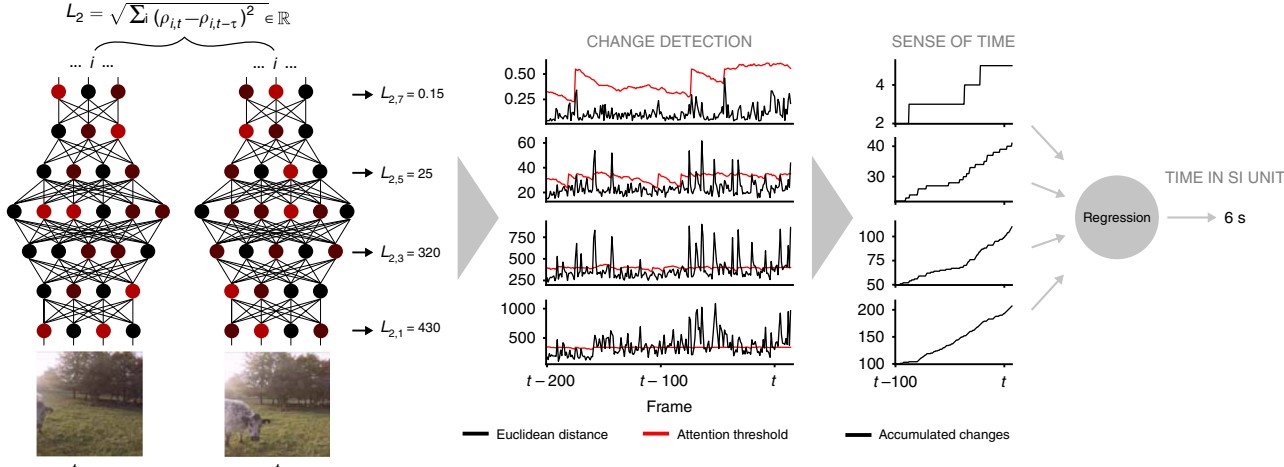

**Fig. 2** Simplified depiction of the time estimation model. Salient changes in network activation driven by video input are accumulated and transformed into standard units for comparison with human reports. The bottom left shows two consecutive frames of video input. The connected coloured nodes depict network structure and activation patterns in each layer in the classification network for the inputs. $L_2$ gives the Euclidean distance between network activations to successive inputs for a given network layer (layers conv2, pool5, fc7, output). Neurons across the hierarchical layers of the classification network are differentially responsive to feature complexity in images, with higher layers more responsive to object-like archetypes and lower layers to primitive features like edges or contours (e.g. see Fig. 4 in ref. [41]). In the Change Detection stage, the value of $L_2$ for a given network layer is compared to a dynamic threshold (red line). When $L_2$ exceeds the threshold level, a salient perceptual change is determined to have occurred, a unit of subjective time is determined to have passed and is accumulated to form the base estimate of time. Support vector regression is applied to convert this abstract time estimate into standard units (in s) for comparison with human reports

Only the pixels of the video inside this spotlight were used as input to the model (see Supplementary Movie 2).

As time estimates generated by the model were made on the same videos as the reports made by humans, human and model estimates could be compared directly. Figure 3a shows duration estimates produced by human participants and the model under the different input scenarios. Participants' reports demonstrated qualities typically found for human estimates of time: over-estimation of short durations and underestimation of long durations (regression of responses to the mean/Vierordt's law), and variance of reports proportional to the reported duration (scalar variability/Weber's law). Model estimates produced when

the full video frame was input (Fig. 3b; Full-frame model) revealed qualitative properties similar to human reports—though the degree of over- and underestimation was exaggerated, the variance of estimates were generally proportional to the estimated duration (see Supplementary Figs. 8 and 9 and Supplementary Discussion: Estimate variance by duration, for detailed exploration of human and model estimate variance). These results demonstrate that the basic method of our model—accumulation of salient changes in activation of a perceptual classification network—can produce meaningful estimates of time. Specifically, the slope of estimation is non-zero with short durations discriminated from long, and the estimates replicate some of

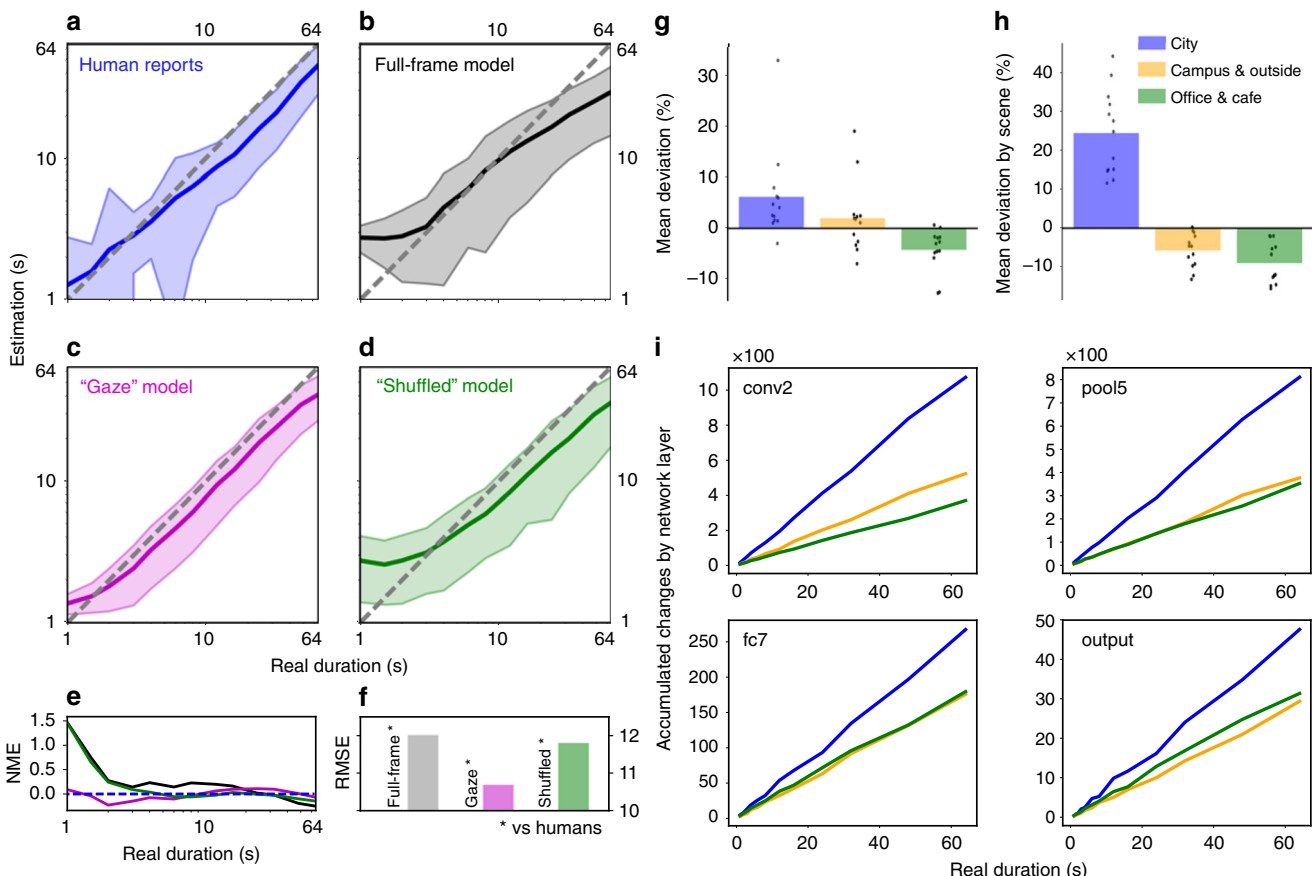

**Fig. 3** Human and model duration estimation and its modulation by scene type. **a** Human duration estimates by duration of presented video between 1 and 64 s (log-log scale). **b**–**d** as for **a** but for model versions Full frame, "Gaze" and "Shuffled". The data in each plot **a**–**d** are based on the same set of 4251 video presentations, reported on by human participants **a** or estimated by different model versions **b**–**d**. Shaded areas in **a**–**d** show ±1 standard deviation of the mean. Human reports in **a** show typical qualities of human temporal estimation with overestimation of short and underestimation of long durations. **b** Model estimates when input the full video frame replicate similar qualitative properties, but estimation was poorer than humans. **c** Model estimates when the input was constrained to approximate human visual-spatial attention, based on human gaze data, very closely approximated human reports made on the same videos. When the gaze contingency was "Shuffled" such that the gaze direction was applied to a different video than that from which it was obtained **d**, performance decreased. **e** Comparison of normalised mean error (NME) in estimations for models and human estimates (normalised by physical duration, across the presented durations). **f** Comparison of the root mean squared error (RMSE) of model estimates compared to the human data. The "Gaze" model is most closely matched (Full frame: 11.99, "Gaze": 10.71, "Shuffled": 11.79). **g** Mean deviation of duration estimates relative to mean estimate by scene type for human participants (mean shown in **a**; City: 6.20 mean (1040 trials), Campus & outside: 2.01 mean (1170 trials), Office & cafe: −4.23 (2080 trials); Total number of trials 4290). **h** As for **g** but for the "Gaze" model (mean shown in **c**; City: 24.41 mean (1035 trials), Campus & outside: −5.75 mean (1167 trials), Office & cafe: −9.00 (2068 trials); Total number of trials 4270). **i** The number of accumulated salient perceptual changes over time in the different network layers (lowest to highest: conv2, pool5, fc7, output) by scene type, for the "Gaze" model shown in **h**

the qualitative aspects of human reports often associated with time perception. However, the overall performance of the system under these conditions still departed from that of human participants (Fig. 3e, f) (see Supplementary Figs. 6 and 7 and Supplementary Discussion: Changes in classification network activation, not just stimulation, are critical to human-like time estimation, for results of experiments conducted on pixel-wise differences in the raw video alone. These data show that tracking changes in classification network activation allows human-like time estimation, while estimation based on tracking changes in the stimulus properties alone does not).

**Human-like gaze improves model performance.** When the video input to the system was constrained to approximate human visual-spatial attention by taking into account gaze position ("Gaze" model; Fig. 3c), model-produced estimates more closely approximated reports made by human participants (Fig. 3c, e, f), with substantially improved estimation as compared to estimates

based on the full frame input. This result was not simply due to the spatial reduction of input caused by the gaze-contingent spatial filtering, nor the movement of the input frame itself, as when the gaze-contingent filtering was applied to videos other than the one from which gaze was recorded (i.e. gaze recorded while viewing one video then applied to a different video; "Shuffled" model), model estimates were poorer (Fig. 3d). These results indicate that the contents of where humans look in a scene play a key role in time perception and indicate that our approach is capturing key features of human time perception, as model performance is improved when input is constrained to be more human-like.

**Model and human time estimation vary by content.** As described in the introduction, human estimates of duration are known to vary by content (e.g. refs. [17,24–27]). In our test videos, three different types of scenes could be broadly identified: scenes filmed moving around a city, moving around a leafy university

campus and surrounding countryside, or from relatively stationary viewpoints inside a cafe or office (Fig. 1d). We reasoned that busy scenes, such as moving around a city, would generally provide more varied perceptual content, with content also being more complex and changing at a faster rate during video presentation. This should mean that city scenes would be judged as longer relative to country/outside and office or cafe scenes. As shown in (Fig. 3g), the pattern of biases in human reports is consistent with this hypothesis. Compared to the global mean estimates (Fig. 3a), reports made about city scenes were judged to be ~6% longer than the mean, while more stationary scenes, such as in a cafe or office, were judged to be ~4% shorter than the overall mean estimation (see Supplementary Fig. 1 for full human and model produced estimates for each tested duration and scene).

To test whether the model-produced estimates exhibited the same content-based biases seen in human duration reports, we examined how model estimates differed by scene type. Following the same reasoning as for the human data, busy city scenes should provide a more varied input, which should lead to more varied activation within the network layers, and therefore greater accumulation of salient changes and a corresponding bias towards overestimation of duration. As shown in Fig. 3h, when the model was shown city scenes, estimates were biased to be longer (~24%) than the overall mean estimation, while estimates for country/outside (~4%) or office/cafe (~7%) scenes were shorter than average. The level of overestimation for city scenes was substantially larger than that found for human reports, but the overall pattern of biases was the same: city > campus/outside > cafe/office. Relative over- and underestimation will partly depend on the content of the training set. In the described experiments, the ratio of scenes containing relatively more changes, such as city and campus or outside scenes was balanced with scenes containing less change, such as office and cafe. Different ratios of training scenes would change the precise over/underestimation, though this is similarly true of human estimation, as the content of previous experience alters subsequent judgements in a number of cases, for example refs. [37,43,44]. See Methods for more details of training and trial composition, and see also Discussion for discussion of system redundancy and its impact on overestimation).

It is important to note again here that the model estimation was not produced based on human estimation data. The support vector regression method mapped accumulated perceptual changes across network layers to the physical durations of the videos, not participant reports. This regression mapping was trained on all video types together, not specifically conducted for each video type separately, meaning that the differences in estimation by scene presented in Fig. 3h reflect the relative differences in the presence of salient perceptual changes in the different scenes (as indicated in Fig. 3i). That the same pattern of biases in estimation is found without explicitly fitting the model to human data indicates the power of the underlying method of accumulating salient changes in perceptual content to produce human-like time perception (see Supplementary Fig. 10 and Supplementary Discussion: Effects of regression on different duration ranges, for results when the regression training is limited to a specific subset of durations, mimicking the block-wise range regression effects often seen in human reports, e.g. refs. [43,44]; Supplementary Fig. 5 and Supplementary Discussion: Model performance is not due to regression overfitting, for evidence against regression overfitting).

Looking into the system performance more deeply, it can be seen that the qualitative matches between human reports and model estimation do not simply arise in the final step of the architecture, wherein the state of the accumulators at each network layer is regressed against physical duration using a support vector regression scheme. Even in the absence of this final step, which transforms accumulated salient changes into standard units of physical time (in s), the system showed the same pattern of biases in accumulation for most durations, at most of the examined network layers. More perceptual changes were accumulated for city scenes than for either of the other scene types, particularly in the lower network layers (conv2 and pool5). Therefore, the regression technique used to transform the tracking of salient perceptual changes is not critical to reproduce these scene-wise biases in duration estimation, and is needed only to compare system performance with human estimation in commensurate units. Indeed humans do not experience time only in seconds; it has been shown that even when they cannot provide a label in seconds, such as in early development, humans can still report about time[45,46]. Learning the mapping between a sensation of time and the associated label in standard units can be considered as a regression problem that needs to be solved during development[45,47]. While regression of accumulated network activation differences into standard units is not critical to reproducing human-like biases in duration perception, basing duration estimation in network activation is key to model performance. When estimates are instead derived directly from differences between successive frames (on a pixel-wise basis) of the video stimuli, bypassing the classification network entirely, generated estimates are substantially worse, and most importantly, do not closely follow human biases by content in estimation. See Supplementary Discussion: Changes in classification network activation, not just stimulation, are critical to human-like time estimation, for more details.

**Accounting for the role of attention in time perception**. The role of attention in human time perception has been extensively studied (see ref. [33] for review). One key finding is that when attention is not specifically directed to tracking time (retrospective time judgements), or actively constrained by other task demands (e.g. high cognitive load), duration estimates differ from when attention is, or can be, freely directed towards time[32–34,36]. Our model is based on detection of salient changes in neural activation underlying perceptual classification. To determine whether a given change is salient, the difference between previous and current network activation is compared to a running threshold, the level of which can be considered to be attention to changes in perceptual classification—effectively attention to time in our conception.

Regarding the influence of the threshold on duration estimation, in our proposal the role of attention to time is intuitive: when the threshold value is high (the red line in Change Detection in Fig. 2 is at a higher level in each layer of the network), a larger difference between successive activations is required in order for a given change to be deemed salient (intuitively, when you are not paying attention to something, you are less likely to notice it changing, but large changes will still be noticed). Consequently, fewer changes in perceptual content are registered within a given epoch and, therefore, duration estimates are shorter. By contrast, when the threshold value is low, smaller differences are deemed salient and more changes are registered, producing generally longer duration estimates. Within our model, it is possible to modulate the level of attention to perceptual change using a single scaling factor referred to as Attention Modulation (see description of Eq. (1)). Changing this scaling factor alters the threshold level, which, following the above description, modulates attention to change in perceptual classification. Shown in Fig. 4 are duration estimates for the "Gaze" model presented in Fig. 3c under different attentional modulation. Lower than normal attention leads to general

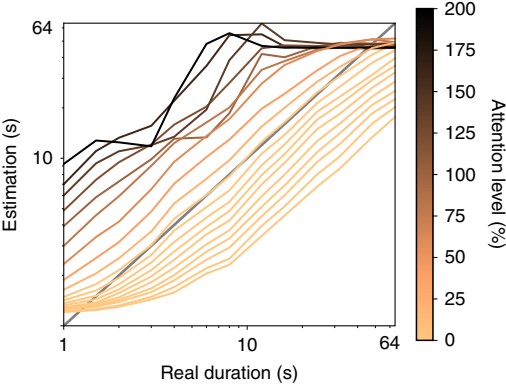

**Fig. 4** Comparison of system duration estimation under different attentional modulation. Attentional modulation refers to a scaling factor applied to the parameters $T_{max}$ and $T_{min}$, specified in Table 1 and Eq. (1). Changing attention affects duration estimates, biasing estimation across a broad range of durations. The model still generally differentiates longer from shorter durations, as indicated by the positive slopes with increasing real duration, but also exhibits biases consistent with those known from behavioural literature associated with attention to time (e.g. refs. [33,34])

underestimation of duration (lighter lines), while higher attention leads to increasing overestimation (darker lines; see Supplementary Fig. 3 for results modulating attention in the other models). Taking duration estimates produced under lower attention and comparing with those produced under higher attention can produce the same pattern of differences in estimation often associated attending (high attention to time; prospective/low cognitive load) or not attending (low attention to time; prospective/high cognitive load) reported in the literature (e.g. see refs. [32–34]). These results demonstrate the flexibility of our model to deal with different demands posed from both "bottom-up" basic stimulus properties as well as "top-down" demands of allocating attention to time.

## Discussion

This study tested the proposal that accumulating salient changes in perceptual content, indicated by differences in successive activations of a perceptual classification network, would be sufficient to produce human-like estimates of duration. Results showed that model-produced estimates could differentiate short from long durations, supporting basic duration estimation. Moreover, when input to the model was constrained to follow human gaze, model estimation improved and became more like human reports. Model estimates were also found to vary by the kind of scene presented, producing the same pattern of biases in estimation seen in human reports for the same videos, with evidence for this bias present even within the accumulation process itself. Finally, we showed that modulating the level of attention to perceptual changes within the proposed model produced systematic under- and overestimation of durations, consistent with previous findings that demonstrate an influence of attention to time on duration estimation. Overall, these results provide compelling support for the hypothesis that human subjective time estimation can be achieved by tracking non-temporal perceptual classification processes, in the absence of any regular pacemaker-like processes.

One might worry that the reliance of our model on a visual classification network is a flaw; after all, it is clear that human time perception depends on more than vision alone, not least because blind people still perceive time. However, the proposal is for a simple conceptual and mechanistic basis to accomplish time

perception under naturalistic conditions using complex stimuli. The model's performance demonstrates the feasibility of this approach when benchmarked against human performance, revealing similar performance under similar constraints. It should be noted that the described human data was obtained with participants seated in a quiet room, and with no auditory track to the video. This created an environment in which the most salient changes during a trial were within the video presentation. Certainly, in an experiment containing audio, audition would contribute to reported duration—and in some cases move human performance away from that of our vision-only model. Similarly, if participants sat in a quiet room with no external stimulation presented, temporal estimations would likely be biased by changes in the internal bodily states of the observer. Indeed, the insula cortex has been suggested to track and accumulate changes in bodily states that contribute to subjective time perception[48,49].

Although the basis of our model is fundamentally visual—an image classification network—similar classification network models exist for audition (e.g. refs. [50,51]), suggesting the possibility to implement the same mechanism in models for auditory classification. This additional level of redundancy in estimation would likely improve performance for scenarios that include both visual and auditory information, as has been shown in other cases where redundant cues from different modalities are combined[23,52–54]. Additional redundancy in estimation would also likely reduce the propensity for the model to overestimate in scenarios that contain many times more perceptual changes than expected (such as indicated by the difference between human and model scene-wise biases Fig. 3h, g). Inclusion of further modules such as memory for previous duration estimations is also likely to improve system estimation as it is now well established that human estimation of duration depends not only on the current experience of a duration, but also past reports of duration[43,44] (see also below discussion of predictive coding). While future extensions of our model could incorporate modules dedicated to auditory, interoceptive, memory and other processes, these possibilities do not detract from the fact that the current implementation provides a simple mechanism that can be applied to these many scenarios, and that when human reports are limited in a similar way to the model, human and model performance are strikingly similar.

The core conception of our proposal shares some similarities with the previously discussed state-dependent network models of time[9,10]. As in our model, the state-dependent network approach suggests that changes in activation patterns within neural networks (network states) over time can be used to estimate time. However, rather than simply saying that any dynamic network has the capacity to represent time by virtue of its changing state, our proposal goes further to say that changes in perceptual classification networks are the basis of content-driven time perception. This position explicitly links time perception and content, and moves away from models of subjective time perception that attempt to track physical time, a notion that has long been identified as conceptually problematic[20]. A primary feature of state-dependent network models is their natural opposition to the classic depiction of time perception as a centralised and unitary process[12,55,56], as suggested in typical pacemaker-based accounts[1–3]. Our suggestion shares this notion of distributed processing, as the information regarding salient changes in perceptual content within a specific modality (vision in this study) is present locally to that modality.

Finally, the described model used Euclidean distance in network activation as the metric of difference between successive inputs—our proxy for perceptual change. While this simple metric was sufficient to deliver a close match between model and human performance, future extensions may consider alternative

metrics. The increasingly influential predictive coding approach to perception[57–61] suggests one such alternative that may increase the explanatory power of the model. Predictive coding accounts are based on the idea that perception is a function of both prediction and current sensory stimulation. Specifically, perceptual content is understood as the brain's "best guess" (Bayesian posterior) of the causes of current sensory input, constrained by prior expectations or predictions. In contrast to bottom-up accounts of perception, in which perceptual content is determined by the hierarchical elaboration of afferent sensory signals, strong predictive coding accounts suggest that bottom-up signals (i.e. flowing from sensory surfaces inwards) carry only the prediction errors (the difference, at each layer in a hierarchy, between actual and predicted signals), with prediction updates passed back down the hierarchy (top-down signals) to inform future perception. A role for predictive coding in time perception has been suggested previously, both in specific[6] and general models[62], and as a general principle to explain behavioural findings[63–67]. Our model exhibits the basic properties of a minimal predictive coding approach; the current network activation state is the best guess (prediction) of the future activation state, and the Euclidean distance between successive activations is the prediction error. The good performance and robustness of our model may reflect this closeness in implementation. While our basic implementation already accounts for some context-based biases in duration estimation (e.g. scene-wise bias), future implementations can include more meaningful "top-down", memory and context-driven constraints on the predicted network state (priors) that will account for a broader range of biases in human estimation.

In summary, subjective time perception is fundamentally related to changes in perceptual content. Here we show that a model built upon detecting salient changes in perceptual content across a hierarchical perceptual classification network can produce human-like time perception for naturalistic stimuli. Critically, model-produced time estimates replicated well-known features of human reports of duration, with estimation differing based on biologically relevant cues, such as where in a scene attention is directed, as well as the general content of a scene (e.g. city or countryside, etc). Moreover, we demonstrated that modulation of the threshold mechanism used to detect salient changes in perceptual content provides the capacity to reproduce the influence of attention to time in duration estimation. That our system produces human-like time estimates based on only natural video inputs, without any appeal to a pacemaker or clock-like mechanism, represents a substantial advance in building artificial systems with human-like temporal cognition, and presents a fresh opportunity to understand human perception and experience of time.

## Methods

**Participants**. Participants were 55 adults (21.2 years, 40 females) recruited from the University of Sussex, participating for course credit or £5 per hour. Participants typically completed 80 trials in the 1 h experimental session, though due to time or other constraints some participants only completed as few as 20 trials (see Data availability statement below for details of how to obtain the raw data, giving the specific trial completion details). Participants provided informed, written consent prior to completing the experiment. This experiment was approved by the University of Sussex ethics committee.

**Apparatus**. Experiments were programmed using Psychtoolbox 3[68] in MATLAB 2012b (MathWorks Inc., Natick, MA, USA) and the Eyelink Toolbox[69], and displayed on a LaCie Electron 22 BLUE II 22" with screen resolution of 1280 × 1024 pixels and refresh rate of 60 Hz. Eye tracking was performed with Eyelink 1000 Plus (SR Research, Mississauga, ON, Canada) at a sampling rate of 1000 Hz, using a desktop camera mount. Head position was stabilised at 57 cm from the screen with a chin and forehead rest.

**Stimuli**. Experimental stimuli were based on videos collected throughout the City of Brighton in the UK, the University of Sussex campus, and the local surrounding area. They were recorded using a GoPro Hero 4 at 60 Hz and 1920 × 1080 pixels, from face height. These videos were processed into candidate stimulus videos 165 min in total duration, at 30 Hz and 1280 × 720 pixels. To create individual trial videos, a pseudo-random list of 4290 trials was generated—330 repetitions of each of 13 durations (1, 1.5, 2, 3, 4, 6, 8, 12, 16, 24, 32, 48 and 64 s). The duration of each trial was pseudo-randomly assigned to the equivalent number of frames in the 165 min of video. There was no attempt to restrict overlap of frames between different trials. The complete trial list is available in the supplied raw data (see Data availability below). Only 4251 of the 4290 total trials were used in the main analyses, as eye-tracking data was missing for technical or other reasons in the excluded trials. In the absence of the eye-tracking data, there was no way to precisely compare the "Gaze" and "Shuffled" models with the human and Full-frame data, so these data were discarded.

For computational experiments when we refer to the "Full Frame" we used the centre 720 × 720 pixel patch from the video (56% of pixels; approximately equivalent to 18 degrees of visual angle (dva) for human observers). When computational experiments used human gaze data, a 400 × 400 pixel patch was centred on the gaze position measured from human participants on that specific trial (about 17% of the image; approximately 10 dva for human observers). The human gaze information was taken from each of the 4290 trials completed by human observers, meaning that the recording of an individual observer's gaze was used for each given simulated trial (rather than an aggregate of many different observers for a single trial).

**Procedure**. Participants typically completed 80 trials in blocks of 20 trials with short periods of rest between blocks. Each block took approximately 12 min to complete. During a block of trials, participants pressed the left mouse key to begin presentation of the video stimulus. Following completion of the video, a visual analogue scale would appear on screen allowing reports of between 0 and 90 s, linearly spaced along the scale. Participants reported the apparent duration of the video by moving the cursor along the visual scale until they were satisfied with their report and confirmed their report by pressing the left mouse button. Participants were instructed to not explicitly count during the stimulus presentations. They were told that counting included physical rhythmic tapping, or the mental equivalent.

**Computational model architecture**. The computational model is made up of four parts: (1) an image classification deep neural network, (2) a threshold mechanism, (3) a set of accumulators and (4) a regression scheme. We used the convolutional deep neural network AlexNet[39] available through the python library caffe[70]. AlexNet had been pre-trained to classify high-resolution images in the LSVRC-2010 ImageNet training set[71] into 1000 different classes, with state-of-the-art performance. It consisted of five convolutional layers, some of which were followed by normalisation and max-pooling layers, and two fully connected layers before the final 1000 class probability output. It has been argued that convolutional networks' connectivity and functionality resemble the connectivity and processing taking place in human visual processing[41] and thus we use this network as the main visual processing system for our computational model. At each time-step (30 Hz), a video frame was fed into the input layer of the network and the subsequent higher layers were activated. For each frame, we extracted the activations of all neurons from layers conv2, pool5, fc7 and the output probabilities. For each layer, we calculated the Euclidean distance between successive states. If the activations were similar, the Euclidean distance would be low, while the distance between neural activations corresponding to frames that include different objects would be high. Each network layer had an initial threshold value for the distance in neural space. This threshold decayed with some stochasticity over time. When the measured Euclidean distance in a layer exceeded the threshold, the counter in that layer's accumulator was incremented by one and the threshold for that layer was reset to its maximum value. Implementation details for each layer can be found in the table below, and the threshold was calculated as:

$$T_{t+1}^k = T_t^k - \left( \frac{T_{max}^k - T_{min}^k}{\tau^k} \right) e^{-\left( \frac{D}{\tau^k} \right)} + \mathcal{N}\left( 0, \frac{T_{max}^k - T_{min}^k}{\alpha} \right), \quad (1)$$

where $T_t^k$ is the threshold value of $k$th layer at timestep $t$ and $D$ indicates the number of timesteps since the last time the threshold value was reset. $T_{max}^k$, $T_{min}^k$

### Table 1 Threshold mechanism parameters

**Parameters for implementing salient event threshold**

| Layer | No. of neurons | $T_{max}$ | $T_{min}$ | $\tau$ | $\alpha$ |
|---|---|---|---|---|---|
| conv2 | 290,400 | 340 | 100 | 100 | 50 |
| pool5 | 9216 | 400 | 100 | 100 | 50 |
| fc7 | 4096 | 35 | 5 | 100 | 50 |
| output | 1000 | 0.55 | 0.15 | 100 | 50 |

and $\tau^k$ are the maximum threshold value, minimum threshold value and decay timeconstant for $k$th layer, respectively, values which are provided in Table 1. Stochastic noise drawn from a Gaussian was added to the threshold and $\alpha$—a dividing constant to adjust the variance of the noise. Finally, the level of attention was modulated by a global scaling factor $C > 0$ applied to the values of $T_{min}^k \leftarrow C \cdot T_{min}^k$ and $T_{max}^k \leftarrow C \cdot T_{max}^k$.

The number of accumulated salient perceptual changes recorded in the accumulators represent the elapsed duration between two points in time. In order to convert estimates of subjective time into units of time in seconds, a simple regression method was used based on epsilon-Support Vector Regression from scikit-learn Python toolkit[72]. The kernel used was the radial basis function with a kernel coefficient of $10^{-4}$ and a penalty parameter for the error term of $10^{-3}$. We used 10-fold cross-validation. To produce the presented data, we used nine out of ten groups for training and one (i.e. 10% of data) for testing. This process was repeated ten times so that each group was used for validation only once.

## Data availability

All data required to reproduce the reported results for both human and model experiments are available at https://doi.org/10.17605/OSF.IO/7UKHJ. The computational model used in this study is available from the Github repository https://github.com/timestorm-project/time-without-clocks.

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

## Acknowledgements

This work was supported by the European Union Future and Emerging Technologies grant (GA:641100) TIMESTORM—Mind and Time: Investigation of the Temporal Traits of Human-Machine Convergence and the Dr. Mortimer and Theresa Sackler Foundation, supporting the Sackler Centre for Consciousness Science. Thanks to Michaela Klimova, Francesca Simonelli and Virginia Mahieu for assistance with the human experiment. Thanks also to Tom Wallis and Andy Philippides for comments on previous versions of the manuscript.

## Author contributions

W.R., D.B., M.S. and A.K.S. conceived of the project. W.R. and D.B. designed initial versions of the reported model, while Z.F., K.N. and W.R. designed the final reported version. Z.F. and K.N. implemented the reported model. W.R., Z.F. and K.N. designed the human experiment and W.R. oversaw data collection. K.N., Z.F. and W.R. analysed the data, prepared the figures and wrote the manuscript. A.K.S., D.B. and M.S. provided critical revisions on the manuscript.

## Additional information

**Competing interests:** The authors declare no competing interests.

