## [Peer Review File · Nature Communications]

Reviewers' comments:

Reviewer #1 (Remarks to the Author):

This paper provides a very interesting perspective on time perception for naturalistic video clips: measuring the amount of "change" over time in a scene can lead to models that match some properties of human time perception. Particularly intriguing was the finding that these matches got better when change was measured for the subvideos that arise from following the trajectory of human observers' eye movements across the "full-frame" video.

One issue that I had while reading this manuscript was related to the full-frame model. If I understood correctly, there is no input of human performance to the model - it exclusively works with the image/video content and the physical durations of the stimuli. Then, however, I was surprised that this model so gravely over- and underestimated short and long videos, respectively. "Neural activation" differences are only taken between pairs of video frames, so in principle the mean duration estimation for each pair of frames should be independent of overall video length, no? In other words, it seems that this effect must be an epiphenomenon of natural video statistics.

Along similar lines, Fig 3E shows the main difference between full-frame and shuffled conditions on the one hand and the gaze-contingent condition on the other hand is most pronounced for short videos (<5-10s). I'm wondering why only short videos are so different. Do the authors have a (tentative) explanation for this effect?

As well, Fig 6 seems to show that there is higher activation at the beginning of the video. Is that an onset effect, or in other words, how was the network initialized before feeding the first frame?

Furthermore, I must admit I found the biological plausibility a bit overstated: the abstract states that [AlexNet] was "functionally similar to human visual processing", l192 claims the proposed model showed "closeness in implementation" to a predictive coder (as in the brain). AlexNet has no temporal knowledge of a /video/ as such - the video and the corresponding output are really just a sequence of individual frames; quite obviously, our brains do not process frames individually (if they did, every eye movement would induce the sensation of drastic change) and there are processes operating at many different time scales.

Minor comments:

Was fixation location in the gaze-contingent condition determined for each observer individually, or was an aggregate used (l63, "the participants' fixation" implies just one position)? If individual fixations were used, this probably means there were 55 times as many simulation runs - could this have had an effect on the SVR (much more training data)? If an aggregate was used, how were fixations aggregated?

I know it's just a schematic figure, but Fig 2 shows a very long time scale (200s), but the longest trials were only 64s; the regressed output then is 200 milliseconds. Is that intentional?

Typos:

l 182, influential a predictive

l 380, 381 psuedo

fig 2 accum//ulated, Eucl/i/d/e/an

Reviewer #2 (Remarks to the Author):

This paper provides a deep network model of timing during the experience of naturalistic sequences of video. The basic idea is that a deep network notices when the features that have been changes in the features it is exposed to, and then a counter is aggregated. This count is then run through an SVM to map onto ratings. The authors claim that the model obeys Veirdodt's law and Weber-law variability. The major claim is that the model's estimate of duration is correlated with humans' ratings and depends on the content of the videos. My biggest problem with this paper is that these findings could be replicated with a really huge range of models, so that this is not even vaguely convincing evidence that a deep network architecture subserves human timing judgements. For instance, how would summed output from a high-pass filter behave? How about we just count the total distance traveled by gaze? Which results would not be predicted? Vierordts' law is predicted by a model that simply guesses an intermediate value for each interval. The mismatch between the complexity of the model and the results being explained is unreasonably high. There is no way we can take these findings as positive evidence for this model beyond the claim that the rate of change of information in the world affects the judgment of time. This is unquestionably true and does not uniquely indicate this general architecture. For instance, suppose that change affects arousal which affects a putative clock.

Specific comments:

* The authors claim that the model exhibits Weber-law error in its performance, as do human observers. I am not convinced by this claim. I know the human literature quite well but I don't see any evidence presented for a good quantitative argument for that in network performance. One might (for instance) evaluate the coefficient of variation and then argue that the CV doesn't change as a function of duration (one then has to argue for the null in a NHST framework). Eyeballing the error bars (under the assumption that they are something like SD--we are not told) this does not seem to be the case in either the judgments or the model ratings. At minimum, there should be some quantitative evaluation of this claim.

* There is a large literature on the cognitive neuroscience of memory judgments and time perception that is ignored here. A recent e-Life paper by Lositsky et al. (2016) is a good starting point. As is the work of Uri Hasson.

* There are two words misspelled in the caption of Figure 2.

Reviewer #3 (Remarks to the Author):

Roseboom et al. propose a hybrid model by which time perception could be accomplished using a non-temporal "feature-extraction" mechanism followed by accumulation of "perceptual saliency". The model consists of three blocks:

- a pre-trained artificial neural system (NN) for classification of images, which extracts a set of features (left panel of Fig.2)
- a set of units that compute activation "changes" or "differences" between the layers of the NN; when activation differences exceed a set of dynamic thresholds, the "salient perceptual changes" are accumulated in a set of accumulators (central panel of Fig.2)
- a support vector machine (SVM) that transforms the pattern of accumulated perceptual changes

into a time duration (right panel of Fig.2).

The interval-timing behavior of the model was contrasted against that of human participants estimating durations from different sets of videos (city, campus&outside, and office&cafe). In accord with known timing phenomena, participants mildly overestimated short durations and mildly underestimated long durations. They also perceived durations to be longer in videos with more changes (city), and perceived durations to be shorter in videos with fewer changes (office&cafe). The "full-frame" model was shown to generate time estimates generally in accord with these biases. The behavior of the model improved when a "gaze" spot (limited input from the full-frame) was implemented.

Based on these findings, the authors infer that the model provides a novel explanation for how human time perception might be accomplished. They argue that their proposed mechanism is novel since it parts with traditional models of timing which mostly use a pacemaker tracking physical time. Although they chose visual modality for implementing their model, they discuss that their idea is a general mechanism which could be applied for different modalities as well (e.g., auditory).

Unfortunately, upon close examination, the "novel" mechanism proposed in this ms is very similar to previously-proposed "multiple scales model" (MTS) (Staddon & Higa, 1999). In short, it is the hierarchical, multiple time scales at which the set of "perceptual saliency" values work that allows the model to capture durations. Since the MTS mechanism can be applied to the output of any dynamic system (e.g., lever-presses generated by a rat, or pixels in a video), the MTS model is most general, and thus the current ms only shows how it can be implemented for image inputs (sets of pixels). Interestingly, the title of the Staddon & Higa 1999 paper is "Time and memory: towards a pacemaker-free theory of interval timing". The MTS model is indeed unanimously considered the first pacemaker-free model of interval timing.

In summary, the idea of perceptual multiple, hierarchically organized temporal scales is already established both in the timing and the visual system literatures, and the current work is only the latest computer implementation of this idea. It is remarkable that the authors re-discovered and applied these ideas to image processing without apparently being aware of Staddon & Higa 1999, although some of Staddon's work was cited. Their work is interesting, but does not raise to the high standards of originality in this journal.

Major concerns

1. Staddon & Higa 1999 MTS core concept is that timing can be captured by a set of hierarchically organized features that characterize the dynamics of the system at different time scales, with lower (faster) units cascading into higher (slower) units. Staddon & Higa showed that the set (pattern) of slow-to-fast features are capable of describing interval-timing (or in other words can identify a specific moment in time). One can easily recognize these ideas in the current work. The pre-trained artificial NN for image classification extracts features (left panel of Fig.2) in each layer, from fast changing (lower layer) to slow changing (higher layers). One can clearly see that the set of perceptual saliency values are distributed at vastly different time scales: lower level saliency accumulates fast (lower graphs of central panel of Fig.2), while higher level saliency accumulates slow (upper graphs of central panel of Fig.2). Crucially, it is the range and hierarchy of time scales of the set of "perceptual saliency" values that allow the model to capture durations, an idea clearly similar to MTS. (The VMS is only a "decoding" mechanism.) This refutes the claim of the authors that the ms describes a "novel" mechanism.

2. Staddon & Higa 1999 seminal paper is entitled "Time and memory: towards a pacemaker-free theory of interval timing". Staddon's work (citations [4] and [5] - although Staddon & Higa 1999 is not cited) is lumped with the "state dependent models". Not all would agree. Citations [4] and [5]

are generally considered, along with Staddon & Higa (1999), as the seminal "pacemaker-free" models. Yet, pacemaker-free models may be considered (as the authors do) as examples of "state-dependent" models. If so, then the current model would also be an example of a "state-dependent" model, and thus its novelty would be equally diminished. This refutes the claim of the authors that the ms describes the first pacemaker-free interval timing model, or that they propose a mechanism radically different from state-dependent models.

3. It is remarkable that the authors re-discovered and applied MTS ideas to image processing, since Staddon & Higa (1999) page 244 also applied their model to the visual system, discussing how their model mimicks the dynamics of the visual system. Staddon & Higa 1999 page 244 cites Glanz 1998 (Science) which discusses Williamson's work (Uusitalo et al. 1996) which indicates that both the visual and auditory systems exhibit a pattern of fast changes in their lower layers (V1) and fast changes in their upper layers (V5). Indeed, the idea of hierarchical perceptual processors with lower-fast layers and upper-slow layers has been further investigated in the literature, see e.g., Hari et al. 2010 Ann.NY Acad Sci: "The brain in time: insights from neuromagnetic recordings" whose abstract states: "The results support the emerging ideas of multiple, hierarchically organized temporal scales in human brain function." This refutes the claim of the authors that the proposed mechanism is new in the visual and auditory systems.

4. Is the proposed computer model truly pacemaker-free? Because the image classification NN functions with inputs that change at a particular frequency, one wonders whether indeed this system is independent of a "frame-based pacemaker". One could argue that it is a simple pacemaker-based model whose pacemaker generates pulses when there is a perceptual saliency change, and that these "changes"/"pulses" accumulate. Within supplemental materials, the authors try to rebut this idea in two ways (1st section of SM). They first show that the behavior of the model is (partly) dependent on content; this is indisputable but it does not demonstrate that the output is independent of the "frame-based pacemaker". Second, authors performed an experiment where the input frequency was changed, supposedly indicating that the model does not depend on frame rate (frequency). Here are a number of supplemental questions that the authors need to address to fully rebut this concern:

- Does the SVM need to be re-trained for each testing condition (city, campus, office, 30Hz, 15Hz, 20% shuffled etc)?
- Does the model work with continuous input, in similarity to the human visual system?
- What is the output of the model when presented with a static image for a set duration? What about a dynamic, noisy pepper-and-salt image?

5. The system is clearly over-fitted and basically can simulate any input-output function. First, the authors re-use a pre-trained image classification NN (citation [38]: Krizhevsky et al. 2012). Krizhevsky et al. state that their system (based on millions of variables - weights) is over-fitted. On top of that, the VMS is also "trained", which adds even more freedom to the system. It is no surprise the system mimics the human behavior (any behavior, really), but it does raise concern whether this is a result of the proposed timing mechanisms or of over-fitting (too many degrees of freedom).

6. Attention is discussed as an independent element which could modulate the system. This adds up yet another degree of freedom (see comment above). This gives the system freedom to adapt to any condition, which makes model so flexible to explain different type of experimental data. That is to say, this gives the model strong predictability but not necessarily makes it a platform to understand true mechanism of human time perception.

7. Time expands when we are waiting for a kettle to boil, but flies when we are having fun. It seems that there is less perceptual saliency in the former case than in the latter. How does the model explain these phenomena? If the proposed explanation involves attention, then the authors need to discuss how attention and perceptual saliency are different and what is the relationship between them?

8. Please add two more panels to SM fig.7, similar to B, for the "Shuffled" and "Full-frame" models.

Minor concerns

- Please clarify how / whether subjects are instructed not to count.
- Please consider using different words for the two "shuffled" conditions / models

Citations

J E Staddon and J J Higa (1999) Time and memory: towards a pacemaker-free theory of interval timing. *J Exp Anal Behav.* 1999 71(2): 215–251. doi: 10.1901/jeab.1999.71-215. PMID: PMC1284701. PMID: 10220931

Glantz J. Magnetic brain imaging traces a stairway to memory. *Science.* 1998 Apr 3;280(5360):37–37

Uusitalo MA1, Williamson SJ, Seppä MT. (1996) Dynamical organisation of the human visual system revealed by lifetimes of activation traces. *Neurosci Lett.* 1996 Aug 9;213(3): 149-52.

Hari R1, Parkkonen L, Nangini C. (2010). The brain in time: insights from neuromagnetic recordings. *Ann N Y Acad Sci.* 2010 Mar;1191:89-109. doi: 10.1111/j.1749-6632.2010.05438.x.

Thanks to the editor and reviewers for their time taken in considering our manuscript. We have made several amendments to the manuscript in response to reviewer comments. However, there are several points made by the reviewers that we find contentious or with which we strongly disagree. Please find below a detailed response to all reviewer comments (replies in bold).

Reviewers' comments:

Reviewer #1 (Remarks to the Author):

This paper provides a very interesting perspective on time perception for naturalistic video clips: measuring the amount of "change" over time in a scene can lead to models that match some properties of human time perception. Particularly intriguing was the finding that these matches got better when change was measured for the subvideos that arise from following the trajectory of human observers' eye movements across the "full-frame" video.

One issue that I had while reading this manuscript was related to the full-frame model. If I understood correctly, there is no input of human performance to the model - it exclusively works with the image/video content and the physical durations of the stimuli. Then, however, I was surprised that this model so gravely over- and underestimated short and long videos, respectively. "Neural activation" differences are only taken between pairs of video frames, so in principle the mean duration estimation for each pair of frames should be independent of overall video length, no? In other words, it seems that this effect must be an epiphenomenon of natural video statistics.

The reviewer is correct in their interpretation that the Full-frame model has no input from human behaviour. The under- and over-estimation is mostly attributable to the regression process that maps the accumulation of salient changes into a labelled estimate in seconds. We have now included additional data in the new version of the manuscript that applies the regression process to several different video duration ranges to demonstrate the effect of regression on estimation – see Supplementary section *Effects of regression on different duration ranges*. These results show that when the regression is conducted on a limited range of durations (e.g. excluding longer durations or shorter durations), you get a range-limited regression to the mean effect, commensurate with that often demonstrated in human behavioural reports (e.g. Jazayeri and Shadlen, 2010, *Nature Neuroscience*; cited in manuscript). These results are flagged in the main text on line 110.

Along similar lines, Fig 3E shows the main difference between full-frame and shuffled conditions on the one hand and the gaze-contingent condition on the other hand is most pronounced for short videos (<5-10s). I'm wondering why only short videos are so different. Do the authors have a (tentative) explanation for this effect?

We think that the difference between the "Gaze" and Full-frame and "Shuffled" estimation at short durations can be attributed to two things: first, the possible variability in changes present to-be-accumulated is higher in the shorter scenes (as videos continue they are more likely to have a mix of rapid or no changes, while shorter videos can be highly biased to extreme cases). Second, in the "Gaze" case, because participants were looking for the part of the scene that does contain change, and finding it, this results in a more reliable, systematic accumulation of changes, wherever they may be present in the scene. By contrast, the Full-frame and "Shuffled" accumulations are highly variable as they don't have this advantage of only being input the informative part of the scene. This makes it difficult for the regression to find the correct mapping between accumulated changes and duration labels for short durations in the Full-frame and "Shuffled" cases – thus the high variability and bias seen in Figure 3E for those model versions.

As well, Fig 6 seems to show that there is higher activation at the beginning of the video. Is that an onset effect, or in other words, how was the network initialized before feeding the first frame?

Just to clarify, Figure 6 shows the *difference* (normalised root-mean squared difference), as percent difference, in accumulation between the normal (30 Hz) and other input frame rates, and so is not directly indicative of activation, per se. What is causing the difference to be “higher” at the short durations is partly dependent on the definition of the measure (making it “higher” rather than “lower”). But as to why there is a difference at all, we think that the large differences are related to both the values, and the range of values, obtained to evaluate the measure at short durations. These values are so small that even small deviations in absolute terms can result in relatively high deviations in the measure (NRMSE (%)) for short compared to longer durations.

Furthermore, I must admit I found the biological plausibility a bit overstated: the abstract states that [AlexNet] was “functionally similar to human visual processing”, l192 claims the proposed model showed “closeness in implementation” to a predictive coder (as in the brain). AlexNet has no temporal knowledge of a /video/ as such - the video and the corresponding output are really just a sequence of individual frames; quite obviously, our brains do not process frames individually (if they did, every eye movement would induce the sensation of drastic change) and there are processes operating at many different time scales.

The debate about biological plausibility of convolutional deep neural nets is very interesting, at least to us, and ongoing. But we would point the reviewer to line 37 of the manuscript where we provide some references in support of our claim of functional similarity. We feel that the language used there “Accumulating evidence supports...” (line 36) is of an appropriate strength in light of the open status of the debate.

Regarding a “Closeness in implementation” to predictive coding – this statement is made in the context of the immediately preceding section where we describe that Euclidean distance in our system is equivalent to a minimal implementation of prediction error (from line 198). AlexNet is indeed static, but the claim that we are making is that differences in successive (static) network activation – again, equivalent to prediction error - may be used as a primary property for duration estimation. This use of prediction error as the basis for operation is the closeness in implementation to which we are referring.

Finally, we would argue that it is well established that our brains do exhibit sampling rates across many levels. The issue of simultaneity is perhaps the oldest in psychology and demonstrates that perception is discrete at different time scales, depending on the perceptual dimension, etc (~8-12 Hz within vision; 2-4 Hz across modality; see VanRullen, 2016, *TICS*; Herzog et al., 2016, *PLOS Biology* for some recent reviews of the issue). One of the great challenges in consciousness science is to understand how smooth and apparently continuous experience results from processing at these different discrete rates.

Minor comments:

Was fixation location in the gaze-contingent condition determined for each observer individually, or was an aggregate used (l63, “the participants’ fixation” implies just one position)? If individual fixations were used, this probably means there were 55 times as many simulation runs - could this have had an effect on the SVR (much more training data)? If an aggregate was used, how were fixations aggregated?

Apologies for the lack of clarity. Each of the 4290 videos was unique (although it could contain overlapping frames – see line 395). Each observer completed 80 trials of this 4290 total (though with some exceptions – noted on line 384). When the input video was modified to track human gaze, we used the values for each individual completed trial. This was not the aggregate of the 55 observers for a given trial, but the specific behaviour for each of the 55 observers for the specific trial that they completed. We have included the following sentence on line 400 in the revised manuscript in order to clarify this issue.

“The gaze information was taken from each of the 4290 trials completed by human observers, meaning that the recording of an individual observer's gaze was used for each given simulated trial (rather than an aggregate of many different observers for a single trial).”

I know it's just a schematic figure, but Fig 2 shows a very long time scale (200s), but the longest trials were only 64s; the regressed output then is 200 milliseconds. Is that intentional?

Sorry for the confusion - our mistake. The x-axis was supposed to be in *frames*, rather than seconds and because it was just a cartoon, we didn't put in the corresponding correct value for the estimation following regression. The figure has now been amended to include both the appropriate label for the x-axis and an appropriate estimation value.

Typos:

l 182, influential a predictive

l 380, 381 psuedo

fig 2 accum//ulated, Eucl/i/d/e/an

Thanks for pointing these out – amended.

Reviewer #2 (Remarks to the Author):

This paper provides a deep network model of timing during the experience of naturalistic sequences of video. The basic idea is that a deep network notices when the features that have been changes in the features it is exposed to, and then a counter is aggregated. This count is then run through an SVM to map onto ratings. The authors claim that the model obeys Veirdodt's law and Weber-law variability. The major claim is that the model's estimate of duration is correlated with humans' ratings and depends on the content of the videos. My biggest problem with this paper is that these findings could be replicated with a really huge range of models, so that this is not even vaguely convincing evidence that a deep network architecture subserves human timing judgements. For instance, how would summed output from a high-pass filter behave? How about we just count the total distance traveled by gaze? Which results would not be predicted? Vierordt's law is predicted by a model that simply guesses an intermediate value for each interval.

The reviewer claims that our model is so trivial as to be equivalent to a variety of extremely simple implementations, including basic gaze statistics and a model that guesses an intermediate level for each trial. In posing this over-simplified position, the reviewer neglects that our model accounts for *all* of these different properties concurrently (that is, Vierordt's law, scalar variability, and matching other biases in human estimates). While each of the approaches suggested by the reviewer may, in themselves, achieve some limited outcomes, none of the suggested approaches can produce the range of properties in human data that are produced by our model. For example, a model that guesses an intermediate value on each trial would produce the same intermediate value regardless of scene type. Alternatively, a model that counts basic gaze statistics would not produce Vierordt's law.

Evidence against these potential criticisms is already included in the manuscript. First, we provide evidence that a 'Gaze' dependent model outperforms a model that includes the same gaze statistics but in a way that is unrelated to the stimulus content (Shuffled model Figure 3D versus Gaze model Figure 3C). We also provide evidence that accumulating only raw differences in the stimulus (world) properties does not account for the differences in estimation by scene type seen in human reports (Supplementary Results section *Changes in classification network activation, not just stimulation, are critical to human-like time estimation*; section title was shown on line 465, now on 492; Supplementary Figure 10). Only when the accumulated differences are viewed through the operation of perceptual classification processes are the produced model estimates qualitatively like human reports. The latter results were previously referred to from line 75 with the following:

“(see Supplemental Results for results of experiments conducted on pixel-wise differences in the raw video alone, by-passing network activation).”

And from line 116:

“See Supplemental Results section Changes in classification network activation, not just stimulation, are critical to human-like time estimation for more details.”

To make these results more apparent to readers, we have changed the first section to the following: (from line 76)

“(see Supplemental Results section *Changes in classification network activation, not just stimulation, are critical to human-like time estimation* for results of experiments conducted on pixel-wise differences in the raw video alone. These data show that tracking changes in classification network activation allows human-like time estimation, while estimation based on tracking changes in the stimulus properties alone does not).”

The mismatch between the complexity of the model and the results being explained is unreasonably high. There is no way we can take these findings as positive evidence for this model beyond the claim that the rate of change of information in the world affects the judgment of time. This is unquestionably true and does not uniquely indicate this general architecture. For instance, suppose that change affects arousal which affects a putative clock.

The aim of the model is to reproduce human-like estimates of duration. Consequently, we compare model estimates against human estimates for the exact same, naturalistic stimuli. If not human reports, it is unclear what the reviewer would admit as evidence in favour of whether a model of human time estimation works.

Over a variety of cases we show that model estimation is well-matched to human reports. Further, we identify specific constraints under which the model performs better. Namely, model performance is improved by constraining input by human gaze (compare Figure 3C and D); the degree to which model estimation matches human data depends on tracking the dynamics of the perceptual classification network, not just rate of change of information in the world (compare Figure 3H against Figure 10, and again refer to Supplemental Results section *Changes in classification network activation, not just stimulation...*). The latter results provide a direct refutation of the reviewer’s statement that: “There is no way we can take these findings as positive evidence for this model beyond the claim that the rate of change of information in the world affects the judgment of time.”

The argument, from parsimony, against an arousal-modulated clock is presented in the paragraph beginning on line 26 as a core motivation for our proposed approach.

Specific comments:

* The authors claim that the model exhibits Weber-law error in its performance, as do human observers. I am not convinced by this claim. I know the human literature quite well but I don’t see any evidence presented for a good quantitative argument for that in network performance. One might (for instance) evaluate the coefficient of variation and then argue that the CV doesn’t change as a function of duration (one then has to argue for the null in a NHST framework). Eyeballing the error bars (under the assumption that they are something like SD---we are not told) this does not seem to be the case in either the judgments or the model ratings. At minimum, there should be some quantitative evaluation of this claim.

The error bars depicted in Figure 3A-D are standard deviation. This is stated in the second line of the Figure 3 caption: “The mean duration estimates for 4290 trials for both human (A) and system (B,C,D) for the range of presented durations (1-64s). Shaded areas show +/-1 standard deviation of the mean.”

As the data in Figure 3A-D are presented on a log-log scale, that the error bar is similar across the entire duration range (Figure 3C in particular) is in support of scalar variability consistent with that often reported for human estimation. This is a minor claim in the manuscript and is only referred to once broadly describing the pattern of results for human (line 68/69) and model (line 70/71) estimation; the broader constellation of findings seems of more interest than this one aspect. However, as further evidence in support of this position, we now also include the coefficient of variation for the four panels depicted in Figure 3A-D. These results are presented in Supplementary Results section *Estimate variance across durations*. The results show that across most tested durations (except those less than ~2 seconds) both human and the different model estimates produce a coefficient of variation that is broadly constant, with the coefficient being approximately 0.5 in all cases. Regression coefficients for linear regressions fitted to the coefficient of variation across duration levels don't provide evidence for a difference from 0-slope for the Human, "Gaze", and "Shuffled" estimates (but do for Full-frame). Overall, these results support the general claim that both our human and model data are consistent with Weber's law/scalar variability. This section is now flagged in the main text:

"(see Supplementary Results section *Estimate variance by duration* for detailed exploration of human and model estimate variance)" on line 71.

* There is a large literature on the cognitive neuroscience of memory judgments and time perception that is ignored here. A recent e-Life paper by Lositsky et al. (2016) is a good starting point. As is the work of Uri Hasson.

In the presented manuscript we don't make explicit reference to the role of narrative or other higher order cognitive effects in time (though we do cite several of the important early relevant works – e.g. Ornstein, 1969; Block, 1974; Block and Reed, 1978; Poynter and Homa, 1983; Poynter, 1989). More importantly, we don't investigate the role of such factors in time perception. Our model makes no attempt to produce or account for cognitive factors beyond basic attention to time. While the work that the reviewer refers to is of definite interest, our model is attempting to characterise and account for the most primary (basic stimulus driven) properties of human time perception, providing a mechanistic basis from which to build and include more complicated processes. We hope to have something more to say about the role of complex, cognitive factors (other than 'attention') in human time perception (and in our model) in the future, but the value of reference to such work in this manuscript, more than we already have, is unclear.

* There are two words misspelled in the caption of Figure 2.

Thanks for pointing these out – they have been amended.

Reviewer #3 (Remarks to the Author):

Roseboom et al. propose a hybrid model by which time perception could be accomplished using a non-temporal "feature-extraction" mechanism followed by accumulation of "perceptual saliency". The model consists of three blocks:

- a pre-trained artificial neural system (NN) for classification of images, which extracts a set of features (left panel of Fig.2)
- a set of units that compute activation "changes" or "differences" between the layers of the NN; when activation differences exceed a set of dynamic thresholds, the "salient perceptual changes" are accumulated in a set of accumulators (central panel of Fig.2)
- a support vector machine (SVM) that transforms the pattern of accumulated perceptual changes into a time duration (right panel of Fig.2).

The interval-timing behavior of the model was contrasted against that of human participants estimating durations from different sets of videos (city, campus&outside, and office&cafe). In accord with known timing phenomena, participants mildly overestimated short durations and mildly underestimated long durations.

They also perceived durations to be longer in videos with more changes (city), and perceived durations to be shorter in videos with fewer changes (office& cafe). The "full-frame" model was shown to generate time estimates generally in accord with these biases. The behavior of the model improved when a "gaze" spot (limited input from the full-frame) was implemented.

Based on these findings, the authors infer that the model provides a novel explanation for how human time perception might be accomplished. They argue that their proposed mechanism is novel since it parts with traditional models of timing which mostly use a pacemaker tracking physical time. Although they chose visual modality for implementing their model, they discuss that their idea is a general mechanism which could be applied for different modalities as well (e.g., auditory).

Unfortunately, upon close examination, the "novel" mechanism proposed in this ms is very similar to previously-proposed "multiple scales model" (MTS) (Staddon & Higa, 1999). In short, it is the hierarchical, multiple time scales at which the set of "perceptual saliency" values work that allows the model to capture durations. Since the MTS mechanism can be applied to the output of any dynamic system (e.g., lever-presses generated by a rat, or pixels in a video), the MTS model is most general, and thus the current ms only shows how it can implemented for image inputs (sets of pixels). Interestingly, the title of the Staddon & Higa 1999 paper is "Time and memory: towards a pacemaker-free theory of interval timing". The MTS model is indeed unanimously considered the first pacemaker-free model of interval timing.

In summary, the idea of perceptual multiple, hierarchically organized temporal scales is already established both in the timing and the visual system literatures, and the current work is only the latest computer implementation of this idea. It is remarkable that the authors re-discovered and applied these ideas to image processing without apparently being aware of Staddon & Higa 1999, although some of Staddon's work was cited. Their work is interesting, but does not raise to the high standards of originality in this journal.

The reviewer's position is dramatically overstated. That time perception relies on multiple time scales and is built out of sequences of events are the only ideas shared between our work and that of Staddon and Higa (1999). The model by Staddon and Higa (1999) is based on memory dynamics over different time scales – seen through the role of habituation. A quick search of our manuscript will find no mention of habituation. Memory in our manuscript is defined only as the accumulation of detected salient changes in perceptual content. There is no change (decay) in these representations once they are accumulated, as would be expected if building a model based on habituation. What our work shows is that the dynamics in perceptual classification (as in our response to reviewer 2, not just the world, compare Figure 3H and Figure 10) provide a sufficient basis for many key aspects of human time perception. We make no claims and present no models regarding the potential role of memory/habituation.

Major concerns

1. Staddon & Higa 1999 MTS core concept is that timing can be captured by a set of hierarchically organized features that characterize the dynamics of the system at different time scales, with lower (faster) units cascading into higher (slower) units. Staddon & Higa showed that the set (pattern) of slow-to-fast features are capable of describing interval-timing (or in other words can identify a specific moment in time). One can easily recognize these ideas in the current work. The pre-trained artificial NN for image classification extracts features (left panel of Fig.2) in each layer, from fast changing (lower layer) to slow changing (higher layers). One can clearly see that the set of perceptual saliency values are distributed at vastly different time scales: lower level saliency accumulates fast (lower graphs of central panel of Fig.2), while higher level saliency accumulates slow (upper graphs of central panel of Fig.2). Crucially, it is the range and hierarchy of time scales of the set of "perceptual saliency" values that allow the model to capture durations, an idea clearly similar to MTS. (The VMS is only a "decoding" mechanism.) This refutes the claim of the authors that the ms describes a "novel" mechanism.

This again is an overstated position. The idea that time is distributed over different scales can be seen in work going back centuries. These concepts can be seen in the posing of time perception as a function of geometric distance of memory by, for example, Hooke (1705)¹. Of interest, Hooke had

already suggested (in the late 17th century) an exponential forgetting function for memory remarkably similar to habituation as modelled by Staddon & Higa (1999)).

It should go without saying that our work builds on prior ideas. We have appropriately cited these ideas where relevant. The common ground with the MTS model is no more than superficial conceptual similarities that are shared between most intuitively meaningful conceptions/models of time.

1. Waller, R. (1705). *The posthumous works of Robert Hooke*, Royal Society, London.

2. Staddon & Higa 1999 seminal paper is entitled "Time and memory: towards a pacemaker-free theory of interval timing". Staddon's work (citations [4] and [5] - although Staddon & Higa 1999 is not cited) is lumped with the "state dependent models". Not all would agree. Citations [4] and [5] are generally considered, along with Staddon & Higa (1999), as the seminal "pacemaker-free" models. Yet, pacemaker-free models may be considered (as the authors do) as examples of "state-dependent" models. If so, then the current model would also be an example of a "state-dependent" model, and thus its novelty would be equally diminished. This refutes the claim of the authors that the ms describes the first pacemaker-free interval timing model, or that they propose a mechanism radically different from state-dependent models.

We cited Staddon (2005) rather than the Staddon and Higa (1999) paper as we wanted to provide a brief overview of several alternative approaches – the review paper by Staddon does a good job of this so that readers may be able to find and interpret the related work easily. These citations are certainly not “lumped in” with the state dependent models. The sentence in which these citations are presented precedes the mention of state dependent models, deliberately placed in contrast. None of the citations at the end of that sentence (line 12) refer to state dependent models. We now include the citation to Staddon and Higa (1999) in addition to the previously cited 2005 review paper – see line 12.

At no point in the manuscript do we assert that our model is the “first pacemaker-free interval timing model”. This is certainly not our claim to novelty. Our claim to novelty is that we can input natural videos, track activity in a model of perceptual classification, produce estimates of duration that vary by scene content, and have the capacity for estimation to be modulated by something conceptually similar to cognitive attention. We would be happy to moderate our claim of novelty if provided with evidence for an alternative approach that achieves this combination of outcomes. Notably, the model presented in Staddon and Higa (1999) and cited by the reviewer does not demonstrate any of these features.

3. It is remarkable that the authors re-discovered and applied MTS ideas to image processing, since Staddon & Higa (1999) page 244 also applied their model to the visual system, discussing how their model mimicks the dynamics of the visual system. Staddon & Higa 1999 page 244 cites Glanz 1998 (Science) which discusses Williamson's work (Uusitalo et al. 1996) which indicates that both the visual and auditory systems exhibit a pattern of fast changes in their lower layers (V1) and fast changes in their upper layers (V5). Indeed, the idea of hierarchical perceptual processors with lower-fast layers and upper-slow layers has been further investigated in the literature, see e.g., Hari et al. 2010 Ann.NY Acad Sci: "The brain in time: insights from neuromagnetic recordings" whose abstract states: "The results support the emerging ideas of multiple, hierarchically organized temporal scales in human brain function." This refutes the claim of the authors that the proposed mechanism is new in the visual and auditory systems.

The above text represents the full extent of the claim by the reviewer for intellectual priority (the full excerpt from Staddon and Higa (1999) is included at the bottom of this letter for verification and context). There is not any more substantive exploration of the issue in the cited paper. In their text, Staddon and Higa (1999) suggest that the dynamics in their model might be detectable in the brain. There is no attempt to explicitly demonstrate that the cited temporal dynamics in brain function relate specifically to the habituation dynamics in their model, only a general gesturing to some neuroimaging work that shows that the brain contains measurable temporal dynamics. To say that this makes our explicit modelling work completely redundant is, we are compelled to say, ridiculous.

Our work uses deep convolutional classification networks as a model for human perceptual classification. This type of network didn't exist when the cited work was produced. Only recently (past 2-3 years) have these kinds of networks been shown to possess the functional properties required for our work to be meaningful (indicated in citations on line 37; a paper out very recently in *Neuron*² furthers our case in the auditory domain) and our work is firmly based in continuing this productive direction of innovative research. To reiterate, our model makes no attempt to model the temporal dynamics of memory, nor how dynamics of memory over different time scales are useful for estimating time (as is the central claim in MTS).

2. Kell, Alexander J.E. and Yamins, Daniel L.K. and Shook, Erica N. and Norman-Haignere, Sam V. and McDermott, Josh H. (2018). A Task-Optimized Neural Network Replicates Human Auditory Behavior, Predicts Brain Responses, and Reveals a Cortical Processing Hierarchy. *Neuron*, 98, 3. doi: 10.1016/j.neuron.2018.03.044

4. Is the proposed computer model truly pacemaker-free? Because the image classification NN functions with inputs that change at a particular frequency, one wonders whether indeed this system is independent of a "frame-based pacemaker". One could argue that it is a simple pacemaker-based model whose pacemaker generates pulses when there is a perceptual saliency change, and that these "changes"/"pulses" accumulate.

Having previously stated that our work is not novel because it simply copies the "first pacemaker free model" of time, the reviewer now contends that our work does depend on a "pacemaker". Evidence against this potential concern is already provided in Supplementary section *Content, not model regularity drives time estimation*.

Within supplemental materials, the authors try to rebut this idea in two ways (1st section of SM). They first show that the behavior of the model is (partly) dependent on content; this is indisputable but it does not demonstrate that the output is independent of the "frame-based pacemaker". Second, authors performed an experiment where the input frequency was changed, supposedly indicating that the model does not depend on frame rate (frequency).

The supplemental result the reviewer refers to (found in Supplementary section *Content, not model regularity drives time estimation*) goes further than to show that estimation is "partly dependent on content" – it shows that leaving the content the same but changing the rate at which the content is updated (sample rate: 30 Hz, 15 Hz or random skipped frames equating to ~24 Hz) does not strongly change accumulation of salient changes in the system. If the sample rate was acting as a pacemaker, as the reviewer suggests, accumulation of changes (the basis of duration estimation) would change proportionally to sample rate. It demonstrably does not (Figure 6). For clarity, we have updated the text in this section to more explicitly state this: (from line 450)

"If the system update rate was simply acting as a pacemaker we would expect that the accumulation of salient perceptual changes would change proportionally to the change in update rate. This was clearly not the case at the basic level of the change accumulation (even before mapping these accumulations into duration labels using support vector regression). Only when the content of the scene was changed, by altering the order in which frames were presented but keeping the standard 30 Hz update rate, was there a large change in accumulated salient perceptual changes. Therefore, these results underline that our system was producing temporal estimates based on the content of the scene, not the update rate of the system."

Here are a number of supplemental questions that the authors need to address to fully rebut this concern:
- Does the SVM need to be re-trained for each testing condition (city, campus, office, 30Hz, 15Hz, 20% shuffled etc)?

As regards the regression training for the different video types (city, campus, office), apologies for not making this clearer. We have now updated the manuscript section that discusses the by-scene differences to the following: (from line 105).

"It is important to note again here that the model estimation was not produced based on human estimation data. The support vector regression method mapped accumulated perceptual changes across network layers to the *physical* durations of the videos, not participant reports. This regression mapping was trained on all video types together, not specifically conducted for each

video type separately, meaning that the differences in estimation by scene presented in Fig. 3H reflect the relative differences in the presence of salient perceptual changes in the different scenes (as indicated in Fig. 3I). That the same pattern of biases in estimation is found without explicitly fitting the model to human data indicates the power of the underlying method of accumulating salient changes in perceptual content to produce human-like time perception.”

For the comparisons of 30 Hz versus 15 Hz, 20% removed frames, and random frame order sequences (as depicted in Figure 6 and Supplementary section *Content, not model regularity drives time estimation*), these comparisons relate to differences in accumulated perceptual changes only, not duration estimations. Consequently, the data presented here did not require any regression training to occur. To make this point clearer, we have updated the text in the section discussing these results: (from line 451)

“This was clearly not the case at the basic level of the change accumulation (even before mapping these accumulations into duration labels using support vector regression).”

And updated the caption for Figure 6 to include:

“Note that the depicted differences are related to raw accumulated perceptual changes, not duration estimation following support vector regression.”

- Does the model work with continuous input, in similarity to the human visual system?

As mentioned in response to reviewer 1, the claim that human perceptual processing does not contain any kind of discrete updating is, at the very least, contentious. As we said to reviewer 1:

We would argue that it is well established that our brains do exhibit sampling rates across many levels. The issue of simultaneity is perhaps the oldest in psychology and demonstrates that perception is discrete at different time scales, depending on the perceptual dimension, etc (~8-12 Hz within vision; 2-4 Hz across modality; see VanRullen, 2016, *TICS*; Herzog et al., 2016, *PLOS Biology* for some recent reviews of the issue). One of the great challenges in consciousness science is to understand how smooth and apparently continuous experience results from processing at these different discrete rates.

- What is the output of the model when presented with a static image for a set duration? What about a dynamic, noisy pepper-and-salt image?

The pattern of results for the suggested inputs is intuitive and simply represents extreme cases of what is already shown in our results – static images would be estimated as shortest and random noise sequences as longest.

Static images, without gaze data, contain no changes and therefore are estimated as shortest for a given duration (there are instances in the Full frame data that approximate the case described by the reviewer).

Random, dynamic (pepper-and-salt) images would produce the opposite to no change, as the reviewer would expect given the paired suggestion. Just as with the data already in the manuscript, videos with the most change (e.g. city scenes) are estimated as the longest. The limitations of the model in this regard are already discussed in the manuscript (from line 171).

While humans would have less trouble with extreme scenarios than our simple model (a model that relies on only a single modality of input and does not have sophisticated memory of past experiences to draw on) this seems a somewhat trivial point to make – humans are massively more complex than our model. Despite this difference in general capability, for the wide variety of stimulation shown to both human participants and our system, estimates are well-matched.

5. The system is clearly over-fitted and basically can simulate any input-output function. First, the authors re-use a pre-trained image classification NN (citation [38]: Krizhevsky et al. 2012). Krizhevsky et al. state that their system (based on millions of variables - weights) is over-fitted. On top of that, the VMS is also

"trained", which adds even more freedom to the system. It is no surprise the system mimics the human behavior (any behavior, really), but it does raise concern whether this is a result of the proposed timing mechanisms or of over-fitting (too many degrees of freedom).

This comment demonstrates a fundamental misunderstanding of our work. The parameters of the perceptual classification network are related to image classification. These are not “free parameters” that could allow us to fit any possible type of data for time estimation. For time estimation, the model contains only three basic parameters at each layer (plus an additional scaling parameter for ‘attention’ where used). These parameters are presented in Equation 1 and Table 1 in the Methods section. Figure 7B demonstrates that the performance of the model(s) is robust over a wide range of values for these few parameters, suggesting that claims of overfitting are unfounded. Further, as presented in the manuscript (Supplementary section *Model performance does not depend on threshold decay*), even when these parameters are simply fixed to a specific value so that threshold level does not change but stays at an intermediate level, model time estimation is still robust (Figure 8).

Regarding potential overfitting for the support vector regression, we provide evidence against this in Supplementary section *Model performance is not due to regression overfitting*. The results presented there demonstrate that overfitting is not a major factor because model performance is only strongly impaired when the regression is trained on very few duration levels (Figure 9). Most importantly, as is stated in the manuscript (e.g. from line 56 and 101 in the previous version, 56 and 105 in the new version), the regression maps the accumulated perceptual changes onto *physical* durations, not human estimates. If model performance is simply due to overfitting it can only be overfitted to the physical durations. Since the model does a poor job of matching physical duration but does a good job of matching the human data in the many ways that we describe, it is unclear to what the reviewer thinks our model is being overfitted.

6. Attention is discussed as an independent element which could modulate the system. This adds up yet another degree of freedom (see comment above). This gives the system freedom to adapt to any condition, which makes model so flexible to explain different type of experimental data. That is to say, this gives the model strong predictability but not necessarily makes it a platform to understand true mechanism of human time perception.

See again the reply to comment 5, but the issue of additional parameters for ‘attention’ also addressed in Supplementary section *Model performance does not depend on threshold decay* and Figure 8. The results presented there show that without allowing the freedom of this ‘attentional’ parameter, the model can still provide good duration estimates (clearly differentiating short from long intervals). The ability to include this parameter, if desired, potentially accounts for a wider variety of behavioural outcomes. This seems to us like an advantage of our approach to be pursued rather than grounds for objection.

7. Time expands when we are waiting for a kettle to boil, but flies when we are having fun. It seems that there is less perceptual saliency in the former case than in the latter. How does the model explain these phenomena? If the proposed explanation involves attention, then the authors need to discuss how attention and perceptual saliency are different and what is the relationship between them?

The suggested aphorisms are intriguing cases, but their relevance here is not entirely clear. We have shown a wide variety of scenes (see Figure 1D) to human participants and obtained their estimates for those scenes. When human participants watched scenes wherein little in the stimulus was changing (such as from a stationary position in an office; putatively a boring scene, comparable to watched-kettles), they reported these scenes as shortest in duration. When they watched busy scenes with many people and changes (such as the city centre scenes; arguably more related to time-flies cases), they reported them as longest. When we then used the exact same scenes as input to our model, it reproduced the same pattern of biases (Figure 3G & H).

As the reviewer mentions, one potential explanation for the difference in experience described by these aphorisms would be in the interaction of stimulation saliency and the degree to which a person attends to time. The reviewer requests an explanation of the relationship between saliency and attention, but our interpretation of this relationship is already given in the section *Accounting for the role of attention in time perception* (from line 129):

“Our model is based on detection of salient changes in neural activation underlying perceptual classification. To determine whether a given change is salient, the difference between previous and current network activation is compared to a running threshold, the level of which can be considered to be attention to changes in perceptual classification – effectively attention to time in our conception. Regarding the influence of the threshold on duration estimation, in our proposal the role of attention to time is intuitive: when the threshold value is high (the red line in Feature Extraction in Fig. 2 is at a higher level in each layer of the network), a larger difference between successive activations is required in order for a given change to be deemed salient (when you aren’t paying attention to something, you are less likely to notice it changing, but large changes will still be noticed). Consequently, fewer changes in perceptual content are registered within a given epoch and, therefore, duration estimates are shorter. By contrast, when the threshold value is low, smaller differences are deemed salient and more changes are registered, producing generally longer duration estimates.”

8. Please add two more panels to SM fig.7, similar to B, for the “Shuffled” and “Full-frame” models.

Amended as suggested.

Minor concerns

- Please clarify how / whether subjects are instructed not to count.

Amended as requested. See line 407.

“Participants were instructed to not explicitly count during the stimulus presentations. They were told that counting included physical rhythmic tapping, or the mental equivalent.”

- Please consider using different words for the two “shuffled” conditions / models

Amended as suggested. The “Shuffled” sequence described in section *Content, not model regularity drives time estimation* and Figure 6 is now referred to as “Random frame order”.

Citations

J E Staddon and J J Higa (1999) Time and memory: towards a pacemaker-free theory of interval timing. *J Exp Anal Behav.* 1999 71(2): 215–251. doi: 10.1901/jeab.1999.71-215. PMID: 10220931

Glantz J. Magnetic brain imaging traces a stairway to memory. *Science.* 1998 Apr 3;280(5360):37–37

Uusitalo MA1, Williamson SJ, Seppä MT. (1996) Dynamical organisation of the human visual system revealed by lifetimes of activation traces. *Neurosci Lett.* 1996 Aug 9;213(3):149-52.

Hari R1, Parkkonen L, Nangini C. (2010). The brain in time: insights from neuromagnetic recordings. *Ann N Y Acad Sci.* 2010 Mar;1191:89-109. doi: 10.1111/j.1749-6632.2010.05438.x.

Here we present the text from Staddon & Higa (1999) on which reviewer 3 bases their claims about novelty. We have marked in bold type some of the more pertinent passages.

Staddon & Higa (1999), page 244-245, *Brain Mechanisms*.

Brain mechanisms.

We believe that behavioural theories stand on their own feet. They are valid to the extent that they describe behavioural data accurately and economically. We argued earlier that given the richness of physiology, the notion of “biological plausibility” is a slippery one. Is a counter and pacemaker more or less plausible than a leaky integrator? Is a system made up of artificial neurons more “physiological” than one composed of thresholds and capacitors? Questions like these seem destined to be inconclusive. All

that really matters in science, we suspect, is how much can be explained with how little (Staddon & Zanutto, 1998). Nevertheless, the pacemaker-accumulator assumptions of SET have inspired a vigorous, and to some degree successful (Gibbon et al., 1997; Meck, 1996), search for underlying physiological mechanisms. It is worth mentioning, therefore, some recent real-time physiological data that seem to fit remarkably closely the basic assumptions of MTS theory. **MTS timing theory is based on five ideas, one about timing and four about habituation: (a) temporal learning uses short-term memory traces as discriminative stimuli; (b) the properties of short-term memory can be understood through the mechanisms of habituation; (c) habituation is a process in which responding is inhibited by a leaky integrator system driven by stimulus input; (d) habituation units are cascaded; and (e) the faster units are on the periphery and the slower ones are further downstream.** In a recent report, Glanz (1998) describes a study reported to the American Physical Society by Williamson and his colleagues that has identified physiological counterparts for the last three assumptions. Williamson's group used a superconducting quantum interference device (SQUID) to detect tiny changes in human brain magnetic activity. Their system recorded maps of whole-brain activity that could be updated every few milliseconds. In the simplest experiment, they looked at brain activity following a single 0.1-s stimulus: "In quick succession, over less than half a second, about a dozen patches lighted up like pinball bumpers, starting with the primary visual cortex in the occipital lobe at the back of the brain" (p. 37). This activation in rapid succession is precisely what we would expect from a series of cascaded units, where the SQUID is detecting changes in V_i , the activation of each integrator. In a second experiment that was in effect a two-trial habituation study with brain activity as the reflex response, subjects were presented twice with a brief (0.1-s) checkerboard stimulus.

They showed the checkerboard twice, with a varying time interval between the displays, to see whether the first stimulus had left any kind of impression along the way. For very brief intervals—10ths of a second—only the areas of initial processing in the back of the brain fired on the second flash, while the others were silent. . . . But as the interval was increased to 10, 20, or even 30 seconds, the downstream areas began firing on the second flash, with a strength finally approaching that of the initial pop. . . . The data imply, says Williamson, that each site has a distinct "forgetting time," ranging from 10ths of a second in the primary visual cortex—the first stage of raw processing—to as long as 30 seconds farther downstream. (p. 37)

Again, this is precisely the behavior of our cascade of habituation units. Because the initial units have fast time constants, they block input to the later, slower units as long as the interstimulus interval is short enough that they have not had time to discharge ("forget") between stimuli; hence, no response of the "downstream" units to the second flash at a short interstimulus interval. But when the interstimulus interval is long, the initial units have already discharged, allowing the stimulus to pass through to later units, which can therefore respond. Williamson continues, "The memories decayed with the simplicity of a capacitor discharging electricity—exponentially with time—and the later an area's place in the processing queue, the longer its memory time was" (p. 37). Apparently brain "memories," like our leaky integrators, forget exponentially. Whether other studies will provide additional physiological counterparts for the MTS theory remains to be seen. But we do believe that the jury is still out on whether pacemaker- accumulator theories or the MTS theory have the stronger claim to biological plausibility.

Reviewers' comments:

Reviewer #1 (Remarks to the Author):

I would like to thank the authors for their responses and clarifications. Just to be clear about what I think must have been a misunderstanding: My concern/criticism that the brain, unlike AlexNet, does not process frames individually, was not meant as a claim that the brain does not do discrete updating. "In isolation" might have been a better choice of words than "individually" - our brains are not simply feed-forward, so temporal context (at different time scales, e.g. adaptation, memory, ...) plays a huge role, and AlexNet knows nothing of that. Certainly, CNNs share some functional similarities with the brain, but so does the telegraph system. Ultimately, however, this discussion might be tangential to the manuscript at hand.

Reviewer #2 (Remarks to the Author):

The model has not changed substantively since the initial submission. I remain unconvinced that it provides novel insight into how humans perceive time. The treatment of cognitive neuroscience and cognitive psychology models is superficial. The data addressed are superficial relative to the body of contemporary cognitive neuroscience and cognitive psychology. The response letter has not seriously engaged with my critique (or the critiques of the other reviewers to be honest).

I do not believe this paper rises to the threshold necessary to justify publication in Nature Communications.

Reviewer #3 (Remarks to the Author):

I am afraid this reviewer remains unconvinced by the rebuttal.

The present model is just too similar to the seminal MTS model of Staddon and Higa (1999) to be considered novel in regard to modeling interval timing. The rebuttal agrees with this point, and steps back the claim of novelty in this regard. Yet, the revised ms continues to underplay the similarity with MTS, and continues to lump MTS - the first pacemaker-free timing model - with other views supposedly "alternative" to pacemakers, of which "state dependent" models are the only ones discussed. The revised ms fails to discuss the conceptual similarity to MTS, although the rebuttal does so at great length. Indeed, the rebuttal - but not the revised ms - agrees that this is not the first pacemaker-free model, and its novelty is not related to being pacemaker-free. This is in stark contrast to the revised ms that continues to imply it is novel in this regard from its very first words: "time without clocks".

The rebuttal contends that the work goes beyond MTS, but the revised ms puts forth no evidence - beyond the ideas previously put forth by MTS - and most critically - as also noted by the other reviewers - puts forth no further biological evidence that their model speaks to fundamental aspects of human time perception or human vision. The revised ms agrees that the present model is only "functionally" related to the visual system, in other words the ms remains in the theoretical realm.

The rebuttal agrees that the sole novelty consists in using the deep-learning networks and video inputs, but albeit novel, these remain simply implementations of the same old (MTS) concepts, not worthy of publication just because they use recent / novel technologies that speak not to the biology of the phenomena. Indeed, novel technologies will continue to evolve and old ideas will continue to be rediscovered over and over again, with little added deep knowledge.

In summary, most of the major concerns raised by this reviewer have been agreed to in the rebuttal, but no true changes were made to the revised ms. The rebuttal agrees with the major concerns that

- this is not the first pacemaker-free timing model,
- the present model builds on previous models, mainly MTS
- no biological evidence has been presented for this model,
- there is only a "functional" semblance to the human visual and/or timing systems.
- the only novel aspect is a the state-of-the-art implementation of same old ideas or models.

With little novelty - only in implementation - the revised ms adds too little to the field to warrant publication in this highly regarded journal.

Thanks to the editor and reviewers for their time taken in considering our manuscript. We have made several amendments to the manuscript in response to reviewer comments. However, there are several points made by the reviewers that we find contentious or with which we strongly disagree. Please find below a detailed response to all reviewer comments (replies in bold).

Reviewers' comments:

Reviewer #1 (Remarks to the Author):

This paper provides a very interesting perspective on time perception for naturalistic video clips: measuring the amount of "change" over time in a scene can lead to models that match some properties of human time perception. Particularly intriguing was the finding that these matches got better when change was measured for the subvideos that arise from following the trajectory of human observers' eye movements across the "full-frame" video.

One issue that I had while reading this manuscript was related to the full-frame model. If I understood correctly, there is no input of human performance to the model - it exclusively works with the image/video content and the physical durations of the stimuli. Then, however, I was surprised that this model so gravely over- and underestimated short and long videos, respectively. "Neural activation" differences are only taken between pairs of video frames, so in principle the mean duration estimation for each pair of frames should be independent of overall video length, no? In other words, it seems that this effect must be an epiphenomenon of natural video statistics.

The reviewer is correct in their interpretation that the Full-frame model has no input from human behaviour. The under- and over-estimation is mostly attributable to the regression process that maps the accumulation of salient changes into a labelled estimate in seconds. We have now included additional data in the new version of the manuscript that applies the regression process to several different video duration ranges to demonstrate the effect of regression on estimation – see Supplementary section *Effects of regression on different duration ranges*. These results show that when the regression is conducted on a limited range of durations (e.g. excluding longer durations or shorter durations), you get a range-limited regression to the mean effect, commensurate with that often demonstrated in human behavioural reports (e.g. Jazayeri and Shadlen, 2010, *Nature Neuroscience*; cited in manuscript). These results are flagged in the main text on line 110.

Along similar lines, Fig 3E shows the main difference between full-frame and shuffled conditions on the one hand and the gaze-contingent condition on the other hand is most pronounced for short videos (<5-10s). I'm wondering why only short videos are so different. Do the authors have a (tentative) explanation for this effect?

We think that the difference between the "Gaze" and Full-frame and "Shuffled" estimation at short durations can be attributed to two things: first, the possible variability in changes present to-be-accumulated is higher in the shorter scenes (as videos continue they are more likely to have a mix of rapid or no changes, while shorter videos can be highly biased to extreme cases). Second, in the "Gaze" case, because participants were looking for the part of the scene that does contain change, and finding it, this results in a more reliable, systematic accumulation of changes, wherever they may be present in the scene. By contrast, the Full-frame and "Shuffled" accumulations are highly variable as they don't have this advantage of only being input the informative part of the scene. This makes it difficult for the regression to find the correct mapping between accumulated changes and duration labels for short durations in the Full-frame and "Shuffled" cases – thus the high variability and bias seen in Figure 3E for those model versions.

As well, Fig 6 seems to show that there is higher activation at the beginning of the video. Is that an onset effect, or in other words, how was the network initialized before feeding the first frame?

Just to clarify, Figure 6 shows the *difference* (normalised root-mean squared difference), as percent difference, in accumulation between the normal (30 Hz) and other input frame rates, and so is not directly indicative of activation, per se. What is causing the difference to be “higher” at the short durations is partly dependent on the definition of the measure (making it “higher” rather than “lower”). But as to why there is a difference at all, we think that the large differences are related to both the values, and the range of values, obtained to evaluate the measure at short durations. These values are so small that even small deviations in absolute terms can result in relatively high deviations in the measure (NRMSE (%)) for short compared to longer durations.

Furthermore, I must admit I found the biological plausibility a bit overstated: the abstract states that [AlexNet] was “functionally similar to human visual processing”, l192 claims the proposed model showed “closeness in implementation” to a predictive coder (as in the brain). AlexNet has no temporal knowledge of a /video/ as such - the video and the corresponding output are really just a sequence of individual frames; quite obviously, our brains do not process frames individually (if they did, every eye movement would induce the sensation of drastic change) and there are processes operating at many different time scales.

The debate about biological plausibility of convolutional deep neural nets is very interesting, at least to us, and ongoing. But we would point the reviewer to line 37 of the manuscript where we provide some references in support of our claim of functional similarity. We feel that the language used there “Accumulating evidence supports...” (line 36) is of an appropriate strength in light of the open status of the debate.

Regarding a “Closeness in implementation” to predictive coding – this statement is made in the context of the immediately preceding section where we describe that Euclidean distance in our system is equivalent to a minimal implementation of prediction error (from line 198). AlexNet is indeed static, but the claim that we are making is that differences in successive (static) network activation – again, equivalent to prediction error - may be used as a primary property for duration estimation. This use of prediction error as the basis for operation is the closeness in implementation to which we are referring.

Finally, we would argue that it is well established that our brains do exhibit sampling rates across many levels. The issue of simultaneity is perhaps the oldest in psychology and demonstrates that perception is discrete at different time scales, depending on the perceptual dimension, etc (~8-12 Hz within vision; 2-4 Hz across modality; see VanRullen, 2016, *TICS*; Herzog et al., 2016, *PLOS Biology* for some recent reviews of the issue). One of the great challenges in consciousness science is to understand how smooth and apparently continuous experience results from processing at these different discrete rates.

Minor comments:

Was fixation location in the gaze-contingent condition determined for each observer individually, or was an aggregate used (l63, “the participants’ fixation” implies just one position)? If individual fixations were used, this probably means there were 55 times as many simulation runs - could this have had an effect on the SVR (much more training data)? If an aggregate was used, how were fixations aggregated?

Apologies for the lack of clarity. Each of the 4290 videos was unique (although it could contain overlapping frames – see line 395). Each observer completed 80 trials of this 4290 total (though with some exceptions – noted on line 384). When the input video was modified to track human gaze, we used the values for each individual completed trial. This was not the aggregate of the 55 observers for a given trial, but the specific behaviour for each of the 55 observers for the specific trial that they completed. We have included the following sentence on line 400 in the revised manuscript in order to clarify this issue.

“The gaze information was taken from each of the 4290 trials completed by human observers, meaning that the recording of an individual observer's gaze was used for each given simulated trial (rather than an aggregate of many different observers for a single trial).”

I know it's just a schematic figure, but Fig 2 shows a very long time scale (200s), but the longest trials were only 64s; the regressed output then is 200 milliseconds. Is that intentional?

Sorry for the confusion - our mistake. The x-axis was supposed to be in *frames*, rather than seconds and because it was just a cartoon, we didn't put in the corresponding correct value for the estimation following regression. The figure has now been amended to include both the appropriate label for the x-axis and an appropriate estimation value.

Typos:

l 182, influential a predictive

l 380, 381 psuedo

fig 2 accum//ulated, Eucl/i/d/e/an

Thanks for pointing these out – amended.

Reviewer #2 (Remarks to the Author):

This paper provides a deep network model of timing during the experience of naturalistic sequences of video. The basic idea is that a deep network notices when the features that have been changes in the features it is exposed to, and then a counter is aggregated. This count is then run through an SVM to map onto ratings. The authors claim that the model obeys Veirdodt's law and Weber-law variability. The major claim is that the model's estimate of duration is correlated with humans' ratings and depends on the content of the videos. My biggest problem with this paper is that these findings could be replicated with a really huge range of models, so that this is not even vaguely convincing evidence that a deep network architecture subserves human timing judgements. For instance, how would summed output from a high-pass filter behave? How about we just count the total distance traveled by gaze? Which results would not be predicted? Vierordt's law is predicted by a model that simply guesses an intermediate value for each interval.

The reviewer claims that our model is so trivial as to be equivalent to a variety of extremely simple implementations, including basic gaze statistics and a model that guesses an intermediate level for each trial. In posing this over-simplified position, the reviewer neglects that our model accounts for *all* of these different properties concurrently (that is, Vierordt's law, scalar variability, and matching other biases in human estimates). While each of the approaches suggested by the reviewer may, in themselves, achieve some limited outcomes, none of the suggested approaches can produce the range of properties in human data that are produced by our model. For example, a model that guesses an intermediate value on each trial would produce the same intermediate value regardless of scene type. Alternatively, a model that counts basic gaze statistics would not produce Vierordt's law.

Evidence against these potential criticisms is already included in the manuscript. First, we provide evidence that a 'Gaze' dependent model outperforms a model that includes the same gaze statistics but in a way that is unrelated to the stimulus content (Shuffled model Figure 3D versus Gaze model Figure 3C). We also provide evidence that accumulating only raw differences in the stimulus (world) properties does not account for the differences in estimation by scene type seen in human reports (Supplementary Results section *Changes in classification network activation, not just stimulation, are critical to human-like time estimation*; section title was shown on line 465, now on 492; Supplementary Figure 10). Only when the accumulated differences are viewed through the operation of perceptual classification processes are the produced model estimates qualitatively like human reports. The latter results were previously referred to from line 75 with the following:

“(see Supplemental Results for results of experiments conducted on pixel-wise differences in the raw video alone, by-passing network activation).”

And from line 116:

“See Supplemental Results section Changes in classification network activation, not just stimulation, are critical to human-like time estimation for more details.”

To make these results more apparent to readers, we have changed the first section to the following: (from line 76)

“(see Supplemental Results section *Changes in classification network activation, not just stimulation, are critical to human-like time estimation* for results of experiments conducted on pixel-wise differences in the raw video alone. These data show that tracking changes in classification network activation allows human-like time estimation, while estimation based on tracking changes in the stimulus properties alone does not).”

The mismatch between the complexity of the model and the results being explained is unreasonably high. There is no way we can take these findings as positive evidence for this model beyond the claim that the rate of change of information in the world affects the judgment of time. This is unquestionably true and does not uniquely indicate this general architecture. For instance, suppose that change affects arousal which affects a putative clock.

The aim of the model is to reproduce human-like estimates of duration. Consequently, we compare model estimates against human estimates for the exact same, naturalistic stimuli. If not human reports, it is unclear what the reviewer would admit as evidence in favour of whether a model of human time estimation works.

Over a variety of cases we show that model estimation is well-matched to human reports. Further, we identify specific constraints under which the model performs better. Namely, model performance is improved by constraining input by human gaze (compare Figure 3C and D); the degree to which model estimation matches human data depends on tracking the dynamics of the perceptual classification network, not just rate of change of information in the world (compare Figure 3H against Figure 10, and again refer to Supplemental Results section *Changes in classification network activation, not just stimulation...*). The latter results provide a direct refutation of the reviewer’s statement that: “There is no way we can take these findings as positive evidence for this model beyond the claim that the rate of change of information in the world affects the judgment of time.”

The argument, from parsimony, against an arousal-modulated clock is presented in the paragraph beginning on line 26 as a core motivation for our proposed approach.

Specific comments:

* The authors claim that the model exhibits Weber-law error in its performance, as do human observers. I am not convinced by this claim. I know the human literature quite well but I don’t see any evidence presented for a good quantitative argument for that in network performance. One might (for instance) evaluate the coefficient of variation and then argue that the CV doesn’t change as a function of duration (one then has to argue for the null in a NHST framework). Eyeballing the error bars (under the assumption that they are something like SD---we are not told) this does not seem to be the case in either the judgments or the model ratings. At minimum, there should be some quantitative evaluation of this claim.

The error bars depicted in Figure 3A-D are standard deviation. This is stated in the second line of the Figure 3 caption: “The mean duration estimates for 4290 trials for both human (A) and system (B,C,D) for the range of presented durations (1-64s). Shaded areas show +/-1 standard deviation of the mean.”

As the data in Figure 3A-D are presented on a log-log scale, that the error bar is similar across the entire duration range (Figure 3C in particular) is in support of scalar variability consistent with that often reported for human estimation. This is a minor claim in the manuscript and is only referred to once broadly describing the pattern of results for human (line 68/69) and model (line 70/71) estimation; the broader constellation of findings seems of more interest than this one aspect. However, as further evidence in support of this position, we now also include the coefficient of variation for the four panels depicted in Figure 3A-D. These results are presented in Supplementary Results section *Estimate variance across durations*. The results show that across most tested durations (except those less than ~2 seconds) both human and the different model estimates produce a coefficient of variation that is broadly constant, with the coefficient being approximately 0.5 in all cases. Regression coefficients for linear regressions fitted to the coefficient of variation across duration levels don't provide evidence for a difference from 0-slope for the Human, "Gaze", and "Shuffled" estimates (but do for Full-frame). Overall, these results support the general claim that both our human and model data are consistent with Weber's law/scalar variability. This section is now flagged in the main text:

"(see Supplementary Results section *Estimate variance by duration* for detailed exploration of human and model estimate variance)" on line 71.

* There is a large literature on the cognitive neuroscience of memory judgments and time perception that is ignored here. A recent e-Life paper by Lositsky et al. (2016) is a good starting point. As is the work of Uri Hasson.

In the presented manuscript we don't make explicit reference to the role of narrative or other higher order cognitive effects in time (though we do cite several of the important early relevant works – e.g. Ornstein, 1969; Block, 1974; Block and Reed, 1978; Poynter and Homa, 1983; Poynter, 1989). More importantly, we don't investigate the role of such factors in time perception. Our model makes no attempt to produce or account for cognitive factors beyond basic attention to time. While the work that the reviewer refers to is of definite interest, our model is attempting to characterise and account for the most primary (basic stimulus driven) properties of human time perception, providing a mechanistic basis from which to build and include more complicated processes. We hope to have something more to say about the role of complex, cognitive factors (other than 'attention') in human time perception (and in our model) in the future, but the value of reference to such work in this manuscript, more than we already have, is unclear.

* There are two words misspelled in the caption of Figure 2.

Thanks for pointing these out – they have been amended.

Reviewer #3 (Remarks to the Author):

Roseboom et al. propose a hybrid model by which time perception could be accomplished using a non-temporal "feature-extraction" mechanism followed by accumulation of "perceptual saliency". The model consists of three blocks:

- a pre-trained artificial neural system (NN) for classification of images, which extracts a set of features (left panel of Fig.2)
- a set of units that compute activation "changes" or "differences" between the layers of the NN; when activation differences exceed a set of dynamic thresholds, the "salient perceptual changes" are accumulated in a set of accumulators (central panel of Fig.2)
- a support vector machine (SVM) that transforms the pattern of accumulated perceptual changes into a time duration (right panel of Fig.2).

The interval-timing behavior of the model was contrasted against that of human participants estimating durations from different sets of videos (city, campus&outside, and office&cafe). In accord with known timing phenomena, participants mildly overestimated short durations and mildly underestimated long durations.

They also perceived durations to be longer in videos with more changes (city), and perceived durations to be shorter in videos with fewer changes (office& cafe). The "full-frame" model was shown to generate time estimates generally in accord with these biases. The behavior of the model improved when a "gaze" spot (limited input from the full-frame) was implemented.

Based on these findings, the authors infer that the model provides a novel explanation for how human time perception might be accomplished. They argue that their proposed mechanism is novel since it parts with traditional models of timing which mostly use a pacemaker tracking physical time. Although they chose visual modality for implementing their model, they discuss that their idea is a general mechanism which could be applied for different modalities as well (e.g., auditory).

Unfortunately, upon close examination, the "novel" mechanism proposed in this ms is very similar to previously-proposed "multiple scales model" (MTS) (Staddon & Higa, 1999). In short, it is the hierarchical, multiple time scales at which the set of "perceptual saliency" values work that allows the model to capture durations. Since the MTS mechanism can be applied to the output of any dynamic system (e.g., lever-presses generated by a rat, or pixels in a video), the MTS model is most general, and thus the current ms only shows how it can implemented for image inputs (sets of pixels). Interestingly, the title of the Staddon & Higa 1999 paper is "Time and memory: towards a pacemaker-free theory of interval timing". The MTS model is indeed unanimously considered the first pacemaker-free model of interval timing.

In summary, the idea of perceptual multiple, hierarchically organized temporal scales is already established both in the timing and the visual system literatures, and the current work is only the latest computer implementation of this idea. It is remarkable that the authors re-discovered and applied these ideas to image processing without apparently being aware of Staddon & Higa 1999, although some of Staddon's work was cited. Their work is interesting, but does not raise to the high standards of originality in this journal.

The reviewer's position is dramatically overstated. That time perception relies on multiple time scales and is built out of sequences of events are the only ideas shared between our work and that of Staddon and Higa (1999). The model by Staddon and Higa (1999) is based on memory dynamics over different time scales – seen through the role of habituation. A quick search of our manuscript will find no mention of habituation. Memory in our manuscript is defined only as the accumulation of detected salient changes in perceptual content. There is no change (decay) in these representations once they are accumulated, as would be expected if building a model based on habituation. What our work shows is that the dynamics in perceptual classification (as in our response to reviewer 2, not just the world, compare Figure 3H and Figure 10) provide a sufficient basis for many key aspects of human time perception. We make no claims and present no models regarding the potential role of memory/habituation.

Major concerns

1. Staddon & Higa 1999 MTS core concept is that timing can be captured by a set of hierarchically organized features that characterize the dynamics of the system at different time scales, with lower (faster) units cascading into higher (slower) units. Staddon & Higa showed that the set (pattern) of slow-to-fast features are capable of describing interval-timing (or in other words can identify a specific moment in time). One can easily recognize these ideas in the current work. The pre-trained artificial NN for image classification extracts features (left panel of Fig.2) in each layer, from fast changing (lower layer) to slow changing (higher layers). One can clearly see that the set of perceptual saliency values are distributed at vastly different time scales: lower level saliency accumulates fast (lower graphs of central panel of Fig.2), while higher level saliency accumulates slow (upper graphs of central panel of Fig.2). Crucially, it is the range and hierarchy of time scales of the set of "perceptual saliency" values that allow the model to capture durations, an idea clearly similar to MTS. (The VMS is only a "decoding" mechanism.) This refutes the claim of the authors that the ms describes a "novel" mechanism.

This again is an overstated position. The idea that time is distributed over different scales can be seen in work going back centuries. These concepts can be seen in the posing of time perception as a function of geometric distance of memory by, for example, Hooke (1705)¹. Of interest, Hooke had

already suggested (in the late 17th century) an exponential forgetting function for memory remarkably similar to habituation as modelled by Staddon & Higa (1999)).

It should go without saying that our work builds on prior ideas. We have appropriately cited these ideas where relevant. The common ground with the MTS model is no more than superficial conceptual similarities that are shared between most intuitively meaningful conceptions/models of time.

1. Waller, R. (1705). *The posthumous works of Robert Hooke*, Royal Society, London.

2. Staddon & Higa 1999 seminal paper is entitled "Time and memory: towards a pacemaker-free theory of interval timing". Staddon's work (citations [4] and [5] - although Staddon & Higa 1999 is not cited) is lumped with the "state dependent models". Not all would agree. Citations [4] and [5] are generally considered, along with Staddon & Higa (1999), as the seminal "pacemaker-free" models. Yet, pacemaker-free models may be considered (as the authors do) as examples of "state-dependent" models. If so, then the current model would also be an example of a "state-dependent" model, and thus its novelty would be equally diminished. This refutes the claim of the authors that the ms describes the first pacemaker-free interval timing model, or that they propose a mechanism radically different from state-dependent models.

We cited Staddon (2005) rather than the Staddon and Higa (1999) paper as we wanted to provide a brief overview of several alternative approaches – the review paper by Staddon does a good job of this so that readers may be able to find and interpret the related work easily. These citations are certainly not “lumped in” with the state dependent models. The sentence in which these citations are presented precedes the mention of state dependent models, deliberately placed in contrast. None of the citations at the end of that sentence (line 12) refer to state dependent models. We now include the citation to Staddon and Higa (1999) in addition to the previously cited 2005 review paper – see line 12.

At no point in the manuscript do we assert that our model is the “first pacemaker-free interval timing model”. This is certainly not our claim to novelty. Our claim to novelty is that we can input natural videos, track activity in a model of perceptual classification, produce estimates of duration that vary by scene content, and have the capacity for estimation to be modulated by something conceptually similar to cognitive attention. We would be happy to moderate our claim of novelty if provided with evidence for an alternative approach that achieves this combination of outcomes. Notably, the model presented in Staddon and Higa (1999) and cited by the reviewer does not demonstrate any of these features.

3. It is remarkable that the authors re-discovered and applied MTS ideas to image processing, since Staddon & Higa (1999) page 244 also applied their model to the visual system, discussing how their model mimicks the dynamics of the visual system. Staddon & Higa 1999 page 244 cites Glanz 1998 (Science) which discusses Williamson's work (Uusitalo et al. 1996) which indicates that both the visual and auditory systems exhibit a pattern of fast changes in their lower layers (V1) and fast changes in their upper layers (V5). Indeed, the idea of hierarchical perceptual processors with lower-fast layers and upper-slow layers has been further investigated in the literature, see e.g., Hari et al. 2010 Ann.NY Acad Sci: "The brain in time: insights from neuromagnetic recordings" whose abstract states: "The results support the emerging ideas of multiple, hierarchically organized temporal scales in human brain function." This refutes the claim of the authors that the proposed mechanism is new in the visual and auditory systems.

The above text represents the full extent of the claim by the reviewer for intellectual priority (the full excerpt from Staddon and Higa (1999) is included at the bottom of this letter for verification and context). There is not any more substantive exploration of the issue in the cited paper. In their text, Staddon and Higa (1999) suggest that the dynamics in their model might be detectable in the brain. There is no attempt to explicitly demonstrate that the cited temporal dynamics in brain function relate specifically to the habituation dynamics in their model, only a general gesturing to some neuroimaging work that shows that the brain contains measurable temporal dynamics. To say that this makes our explicit modelling work completely redundant is, we are compelled to say, ridiculous.

Our work uses deep convolutional classification networks as a model for human perceptual classification. This type of network didn't exist when the cited work was produced. Only recently (past 2-3 years) have these kinds of networks been shown to possess the functional properties required for our work to be meaningful (indicated in citations on line 37; a paper out very recently in *Neuron*² furthers our case in the auditory domain) and our work is firmly based in continuing this productive direction of innovative research. To reiterate, our model makes no attempt to model the temporal dynamics of memory, nor how dynamics of memory over different time scales are useful for estimating time (as is the central claim in MTS).

2. Kell, Alexander J.E. and Yamins, Daniel L.K. and Shook, Erica N. and Norman-Haignere, Sam V. and McDermott, Josh H. (2018). A Task-Optimized Neural Network Replicates Human Auditory Behavior, Predicts Brain Responses, and Reveals a Cortical Processing Hierarchy. *Neuron*, 98, 3. doi: 10.1016/j.neuron.2018.03.044

4. Is the proposed computer model truly pacemaker-free? Because the image classification NN functions with inputs that change at a particular frequency, one wonders whether indeed this system is independent of a "frame-based pacemaker". One could argue that it is a simple pacemaker-based model whose pacemaker generates pulses when there is a perceptual saliency change, and that these "changes"/"pulses" accumulate.

Having previously stated that our work is not novel because it simply copies the "first pacemaker free model" of time, the reviewer now contends that our work does depend on a "pacemaker". Evidence against this potential concern is already provided in Supplementary section *Content, not model regularity drives time estimation*.

Within supplemental materials, the authors try to rebut this idea in two ways (1st section of SM). They first show that the behavior of the model is (partly) dependent on content; this is indisputable but it does not demonstrate that the output is independent of the "frame-based pacemaker". Second, authors performed an experiment where the input frequency was changed, supposedly indicating that the model does not depend on frame rate (frequency).

The supplemental result the reviewer refers to (found in Supplementary section *Content, not model regularity drives time estimation*) goes further than to show that estimation is "partly dependent on content" – it shows that leaving the content the same but changing the rate at which the content is updated (sample rate: 30 Hz, 15 Hz or random skipped frames equating to ~24 Hz) does not strongly change accumulation of salient changes in the system. If the sample rate was acting as a pacemaker, as the reviewer suggests, accumulation of changes (the basis of duration estimation) would change proportionally to sample rate. It demonstrably does not (Figure 6). For clarity, we have updated the text in this section to more explicitly state this: (from line 450)

"If the system update rate was simply acting as a pacemaker we would expect that the accumulation of salient perceptual changes would change proportionally to the change in update rate. This was clearly not the case at the basic level of the change accumulation (even before mapping these accumulations into duration labels using support vector regression). Only when the content of the scene was changed, by altering the order in which frames were presented but keeping the standard 30 Hz update rate, was there a large change in accumulated salient perceptual changes. Therefore, these results underline that our system was producing temporal estimates based on the content of the scene, not the update rate of the system."

Here are a number of supplemental questions that the authors need to address to fully rebut this concern:
- Does the SVM need to be re-trained for each testing condition (city, campus, office, 30Hz, 15Hz, 20% shuffled etc)?

As regards the regression training for the different video types (city, campus, office), apologies for not making this clearer. We have now updated the manuscript section that discusses the by-scene differences to the following: (from line 105).

"It is important to note again here that the model estimation was not produced based on human estimation data. The support vector regression method mapped accumulated perceptual changes across network layers to the *physical* durations of the videos, not participant reports. This regression mapping was trained on all video types together, not specifically conducted for each

video type separately, meaning that the differences in estimation by scene presented in Fig. 3H reflect the relative differences in the presence of salient perceptual changes in the different scenes (as indicated in Fig. 3I). That the same pattern of biases in estimation is found without explicitly fitting the model to human data indicates the power of the underlying method of accumulating salient changes in perceptual content to produce human-like time perception.”

For the comparisons of 30 Hz versus 15 Hz, 20% removed frames, and random frame order sequences (as depicted in Figure 6 and Supplementary section *Content, not model regularity drives time estimation*), these comparisons relate to differences in accumulated perceptual changes only, not duration estimations. Consequently, the data presented here did not require any regression training to occur. To make this point clearer, we have updated the text in the section discussing these results: (from line 451)

“This was clearly not the case at the basic level of the change accumulation (even before mapping these accumulations into duration labels using support vector regression).”

And updated the caption for Figure 6 to include:

“Note that the depicted differences are related to raw accumulated perceptual changes, not duration estimation following support vector regression.”

- Does the model work with continuous input, in similarity to the human visual system?

As mentioned in response to reviewer 1, the claim that human perceptual processing does not contain any kind of discrete updating is, at the very least, contentious. As we said to reviewer 1:

We would argue that it is well established that our brains do exhibit sampling rates across many levels. The issue of simultaneity is perhaps the oldest in psychology and demonstrates that perception is discrete at different time scales, depending on the perceptual dimension, etc (~8-12 Hz within vision; 2-4 Hz across modality; see VanRullen, 2016, *TICS*; Herzog et al., 2016, *PLOS Biology* for some recent reviews of the issue). One of the great challenges in consciousness science is to understand how smooth and apparently continuous experience results from processing at these different discrete rates.

- What is the output of the model when presented with a static image for a set duration? What about a dynamic, noisy pepper-and-salt image?

The pattern of results for the suggested inputs is intuitive and simply represents extreme cases of what is already shown in our results – static images would be estimated as shortest and random noise sequences as longest.

Static images, without gaze data, contain no changes and therefore are estimated as shortest for a given duration (there are instances in the Full frame data that approximate the case described by the reviewer).

Random, dynamic (pepper-and-salt) images would produce the opposite to no change, as the reviewer would expect given the paired suggestion. Just as with the data already in the manuscript, videos with the most change (e.g. city scenes) are estimated as the longest. The limitations of the model in this regard are already discussed in the manuscript (from line 171).

While humans would have less trouble with extreme scenarios than our simple model (a model that relies on only a single modality of input and does not have sophisticated memory of past experiences to draw on) this seems a somewhat trivial point to make – humans are massively more complex than our model. Despite this difference in general capability, for the wide variety of stimulation shown to both human participants and our system, estimates are well-matched.

5. The system is clearly over-fitted and basically can simulate any input-output function. First, the authors re-use a pre-trained image classification NN (citation [38]: Krizhevsky et al. 2012). Krizhevsky et al. state that their system (based on millions of variables - weights) is over-fitted. On top of that, the VMS is also

"trained", which adds even more freedom to the system. It is no surprise the system mimics the human behavior (any behavior, really), but it does raise concern whether this is a result of the proposed timing mechanisms or of over-fitting (too many degrees of freedom).

This comment demonstrates a fundamental misunderstanding of our work. The parameters of the perceptual classification network are related to image classification. These are not “free parameters” that could allow us to fit any possible type of data for time estimation. For time estimation, the model contains only three basic parameters at each layer (plus an additional scaling parameter for ‘attention’ where used). These parameters are presented in Equation 1 and Table 1 in the Methods section. Figure 7B demonstrates that the performance of the model(s) is robust over a wide range of values for these few parameters, suggesting that claims of overfitting are unfounded. Further, as presented in the manuscript (Supplementary section *Model performance does not depend on threshold decay*), even when these parameters are simply fixed to a specific value so that threshold level does not change but stays at an intermediate level, model time estimation is still robust (Figure 8).

Regarding potential overfitting for the support vector regression, we provide evidence against this in Supplementary section *Model performance is not due to regression overfitting*. The results presented there demonstrate that overfitting is not a major factor because model performance is only strongly impaired when the regression is trained on very few duration levels (Figure 9). Most importantly, as is stated in the manuscript (e.g. from line 56 and 101 in the previous version, 56 and 105 in the new version), the regression maps the accumulated perceptual changes onto *physical* durations, not human estimates. If model performance is simply due to overfitting it can only be overfitted to the physical durations. Since the model does a poor job of matching physical duration but does a good job of matching the human data in the many ways that we describe, it is unclear to what the reviewer thinks our model is being overfitted.

6. Attention is discussed as an independent element which could modulate the system. This adds up yet another degree of freedom (see comment above). This gives the system freedom to adapt to any condition, which makes model so flexible to explain different type of experimental data. That is to say, this gives the model strong predictability but not necessarily makes it a platform to understand true mechanism of human time perception.

See again the reply to comment 5, but the issue of additional parameters for ‘attention’ also addressed in Supplementary section *Model performance does not depend on threshold decay* and Figure 8. The results presented there show that without allowing the freedom of this ‘attentional’ parameter, the model can still provide good duration estimates (clearly differentiating short from long intervals). The ability to include this parameter, if desired, potentially accounts for a wider variety of behavioural outcomes. This seems to us like an advantage of our approach to be pursued rather than grounds for objection.

7. Time expands when we are waiting for a kettle to boil, but flies when we are having fun. It seems that there is less perceptual saliency in the former case than in the latter. How does the model explain these phenomena? If the proposed explanation involves attention, then the authors need to discuss how attention and perceptual saliency are different and what is the relationship between them?

The suggested aphorisms are intriguing cases, but their relevance here is not entirely clear. We have shown a wide variety of scenes (see Figure 1D) to human participants and obtained their estimates for those scenes. When human participants watched scenes wherein little in the stimulus was changing (such as from a stationary position in an office; putatively a boring scene, comparable to watched-kettles), they reported these scenes as shortest in duration. When they watched busy scenes with many people and changes (such as the city centre scenes; arguably more related to time-flies cases), they reported them as longest. When we then used the exact same scenes as input to our model, it reproduced the same pattern of biases (Figure 3G & H).

As the reviewer mentions, one potential explanation for the difference in experience described by these aphorisms would be in the interaction of stimulation saliency and the degree to which a person attends to time. The reviewer requests an explanation of the relationship between saliency and attention, but our interpretation of this relationship is already given in the section *Accounting for the role of attention in time perception* (from line 129):

“Our model is based on detection of salient changes in neural activation underlying perceptual classification. To determine whether a given change is salient, the difference between previous and current network activation is compared to a running threshold, the level of which can be considered to be attention to changes in perceptual classification – effectively attention to time in our conception. Regarding the influence of the threshold on duration estimation, in our proposal the role of attention to time is intuitive: when the threshold value is high (the red line in Feature Extraction in Fig. 2 is at a higher level in each layer of the network), a larger difference between successive activations is required in order for a given change to be deemed salient (when you aren’t paying attention to something, you are less likely to notice it changing, but large changes will still be noticed). Consequently, fewer changes in perceptual content are registered within a given epoch and, therefore, duration estimates are shorter. By contrast, when the threshold value is low, smaller differences are deemed salient and more changes are registered, producing generally longer duration estimates.”

8. Please add two more panels to SM fig.7, similar to B, for the “Shuffled” and “Full-frame” models.

Amended as suggested.

Minor concerns

- Please clarify how / whether subjects are instructed not to count.

Amended as requested. See line 407.

“Participants were instructed to not explicitly count during the stimulus presentations. They were told that counting included physical rhythmic tapping, or the mental equivalent.”

- Please consider using different words for the two “shuffled” conditions / models

Amended as suggested. The “Shuffled” sequence described in section *Content, not model regularity drives time estimation* and Figure 6 is now referred to as “Random frame order”.

Citations

J E Staddon and J J Higa (1999) Time and memory: towards a pacemaker-free theory of interval timing. J Exp Anal Behav. 1999 71(2): 215–251. doi: 10.1901/jeab.1999.71-215. PMID: 10220931

Glantz J. Magnetic brain imaging traces a stairway to memory. Science. 1998 Apr 3;280(5360):37–37

Uusitalo MA1, Williamson SJ, Seppä MT. (1996) Dynamical organisation of the human visual system revealed by lifetimes of activation traces. Neurosci Lett. 1996 Aug 9;213(3):149-52.

Hari R1, Parkkonen L, Nangini C. (2010). The brain in time: insights from neuromagnetic recordings. Ann N Y Acad Sci. 2010 Mar;1191:89-109. doi: 10.1111/j.1749-6632.2010.05438.x.

Here we present the text from Staddon & Higa (1999) on which reviewer 3 bases their claims about novelty. We have marked in bold type some of the more pertinent passages.

Staddon & Higa (1999), page 244-245, *Brain Mechanisms*.

Brain mechanisms.

We believe that behavioural theories stand on their own feet. They are valid to the extent that they describe behavioural data accurately and economically. We argued earlier that given the richness of physiology, the notion of “biological plausibility” is a slippery one. Is a counter and pacemaker more or less plausible than a leaky integrator? Is a system made up of artificial neurons more “physiological” than one composed of thresholds and capacitors? Questions like these seem destined to be inconclusive. All

that really matters in science, we suspect, is how much can be explained with how little (Staddon & Zanutto, 1998). Nevertheless, the pacemaker-accumulator assumptions of SET have inspired a vigorous, and to some degree successful (Gibbon et al., 1997; Meck, 1996), search for underlying physiological mechanisms. It is worth mentioning, therefore, some recent real-time physiological data that seem to fit remarkably closely the basic assumptions of MTS theory. **MTS timing theory is based on five ideas, one about timing and four about habituation: (a) temporal learning uses short-term memory traces as discriminative stimuli; (b) the properties of short-term memory can be understood through the mechanisms of habituation; (c) habituation is a process in which responding is inhibited by a leaky integrator system driven by stimulus input; (d) habituation units are cascaded; and (e) the faster units are on the periphery and the slower ones are further downstream.** In a recent report, Glanz (1998) describes a study reported to the American Physical Society by Williamson and his colleagues that has identified physiological counterparts for the last three assumptions. Williamson's group used a superconducting quantum interference device (SQUID) to detect tiny changes in human brain magnetic activity. Their system recorded maps of whole-brain activity that could be updated every few milliseconds. In the simplest experiment, they looked at brain activity following a single 0.1-s stimulus: "In quick succession, over less than half a second, about a dozen patches lighted up like pinball bumpers, starting with the primary visual cortex in the occipital lobe at the back of the brain" (p. 37). This activation in rapid succession is precisely what we would expect from a series of cascaded units, where the SQUID is detecting changes in V_i , the activation of each integrator. In a second experiment that was in effect a two-trial habituation study with brain activity as the reflex response, subjects were presented twice with a brief (0.1-s) checkerboard stimulus.

They showed the checkerboard twice, with a varying time interval between the displays, to see whether the first stimulus had left any kind of impression along the way. For very brief intervals—10ths of a second—only the areas of initial processing in the back of the brain fired on the second flash, while the others were silent. . . . But as the interval was increased to 10, 20, or even 30 seconds, the downstream areas began firing on the second flash, with a strength finally approaching that of the initial pop. . . . The data imply, says Williamson, that each site has a distinct "forgetting time," ranging from 10ths of a second in the primary visual cortex—the first stage of raw processing—to as long as 30 seconds farther downstream. (p. 37)

Again, this is precisely the behavior of our cascade of habituation units. Because the initial units have fast time constants, they block input to the later, slower units as long as the interstimulus interval is short enough that they have not had time to discharge ("forget") between stimuli; hence, no response of the "downstream" units to the second flash at a short interstimulus interval. But when the interstimulus interval is long, the initial units have already discharged, allowing the stimulus to pass through to later units, which can therefore respond. Williamson continues, "The memories decayed with the simplicity of a capacitor discharging electricity—exponentially with time—and the later an area's place in the processing queue, the longer its memory time was" (p. 37). Apparently brain "memories," like our leaky integrators, forget exponentially. Whether other studies will provide additional physiological counterparts for the MTS theory remains to be seen. But we do believe that the jury is still out on whether pacemaker- accumulator theories or the MTS theory have the stronger claim to biological plausibility.